# OmniCorpus: A Unified Multimodal Corpus of 10 Billion-Level Images Interleaved with Text

**Qingyun Li**[2,1*], **Zhe Chen**[3,1*], **Weiyun Wang**[4,1*], **Wenhai Wang**[5,1*], **Shenglong Ye**[1*],
**Zhenjiang Jin**[1*], **Guanzhou Chen**[1*], **Yinan He**[1*], **Zhangwei Gao**[1*], **Erfei Cui**[1*],
**Jiashuo Yu**[1*], **Hao Tian**[6*], **Jiasheng Zhou**[6*], **Chao Xu**[1*], **Bin Wang**[1*], **Xingjian Wei**[1*],
**Wei Li**[1*], **Wenjian Zhang**[1*], **Bo Zhang**[1*], **Pinlong Cai**[1*], **Licheng Wen**[1*], **Xiangchao Yan**[1*],
**Pei Chu**[1*], **Yi Wang**[1*], **Min Dou**[1], **Changyao Tian**[5,1], **Xizhou Zhu**[6,1,7], **Lewei Lu**[6],
**Yushi Chen**[2], **Junjun He**[1], **Tong Lu**[3], **Yali Wang**[1], **Limin Wang**[3,1], **Dahua Lin**[1],
**Yu Qiao**[1], **Botian Shi**[1], **Conghui He**[1✉], **Jifeng Dai**[7,1✉]

[1]Shanghai AI Laboratory, [2]Harbin Institute of Technology, [3]Nanjing University,
[4]Fudan University, [5]The Chinese University of Hong Kong, [6]SenseTime Research,
[7]Tsinghua University,

## Abstract

Image-text interleaved data, consisting of multiple images and texts arranged in a natural document format, aligns with the presentation paradigm of internet data and closely resembles human reading habits. Recent studies have shown that such data aids multimodal in-context learning and maintains the capabilities of large language models during multimodal fine-tuning. However, the limited scale and diversity of current image-text interleaved data restrict the development of multimodal large language models. In this paper, we introduce OmniCorpus, a 10 billion-level open-source image-text interleaved dataset. Using an efficient data engine, we filter and extract large-scale high-quality documents, which contain 8.6 billion images and 1,696 billion text tokens. Compared to counterparts (*e.g.*, MMC4, OBELICS), our dataset 1) has 15 times larger scales while maintaining good data quality; 2) features more diverse sources, including both English and non-English websites as well as video-centric websites; 3) is more flexible, easily degradable from an image-text interleaved format to pure text corpus and image-text pairs. Through comprehensive analysis and experiments, we validate the quality, usability, and effectiveness of the proposed dataset. We hope this could provide a solid data foundation for future multimodal model research. Code and data are released at https://github.com/OpenGVLab/OmniCorpus.

## 1 Introduction

With the rise of large language models (LLMs) (Zheng et al., 2024; Team, 2023; Cai et al., 2024; Bai et al., 2023a; Touvron et al., 2023a;b; Bi et al., 2024; Brown et al., 2020; Achiam et al., 2023; Zeng et al., 2022), multimodal large language models (MLLMs) (OpenAI, 2023; Liu et al., 2023e;d; Chen et al., 2023b; 2024b; Bai et al., 2023b; Team et al., 2023; Reid et al., 2024; Zhu et al., 2023a; Alayrac et al., 2022; Sun et al., 2023c; Ge et al., 2024) have also made significant progress. These MLLMs typically integrate pre-trained LLMs with vision foundation models (VFMs) (Radford et al., 2021; Ilharco et al., 2021; Chen et al., 2023b; Zhai et al., 2023; Sun et al., 2023b), aligning them through extensive image-text pairing datasets (*e.g.*, LAION (Schuhmann et al., 2022) and COYO (Byeon et al., 2022)), thereby enabling the comprehension of visual cues within language models. These datasets, collected by web scraping to match images with their descriptive captions, establish robust links between visual and linguistic elements. Nonetheless, they neglect the original structure of documents, leading to a loss of contextual details and resulting in lower text quality and lack of contextual richness compared to the training corpus of LLMs. Therefore, there is an imperative need

*to investigate more natural and flexible multimodal data that go beyond naive image-text pairings, with the aim of enhancing the training efficacy of MLLMs.*

Pioneering studies (Zhu et al., 2024; Laurençon et al., 2024a; McKinzie et al., 2024; Alayrac et al., 2022) have introduced image-text interleaved data, demonstrating their promise in preserving the linguistic prowess of LLMs and boosting few-shot capabilities in tasks such as image captioning and visual question answering (VQA). Despite this progress, the scale of these datasets remains relatively limited, with the most extensive containing approximately 140 million documents, significantly smaller than well-established text or image-text pair datasets. Moreover, their primary data sources, mostly English websites from Common Crawl (CC) (Common Crawl, 2007), restrict content variety. These constraints hinder the datasets' capacity to fully unleash the potential of MLLMs, restricting their advancement and performance.

Given these considerations, constructing large-scale high-quality image-text interleaved data for MLLMs involves addressing several key challenges: (1) *Diverse data sources:* existing sources like CC are relatively homogeneous, which are mainly text-centric with few images. In addition, the availability of CC images is nearing exhaustion, making it difficult to support the scaling up of future multimodal models. (2) *Large-scale data processing:* An efficient, scalable, and parallelizable data engine is required to handle the massive volumes of multimodal data involved in this task. (3) *High-quality multimodal data:* Comprehensive image and text filters are also crucial to ensure that the generated text corpus maintains the same high quality as the original training data of LLMs while interleaving high-quality images.

In this work, to establish a solid data foundation for MLLM research, we introduce OmniCorpus, a 10 billion-level open-source image-text interleaved dataset. To expand data sources and address the exhaustion of CC images, we supplement our dataset with data from non-English websites and high-quality image content from video platforms. We propose a unified data format, termed streaming data format, which is not only flexible to store image and text data from different sources, but also facilitates subsequent data reading, visualization, and data cleaning. To efficiently leverage the large-scale data from multiple sources, we develop *an efficient data pipeline capable of scaling to thousands of CPU cores*. We carefully review the overall pipeline of the data engine and optimize each component (*e.g.*, main body extraction, preliminary text filtering) for higher efficiency and speedup ratio in a parallel framework. To enhance data quality, we implement a *human-feedback text filter* to reduce the noise within the texts, such as advertisements and other irrelevant content.

As shown in Figure 1 and Table 1, our OmniCorpus dataset demonstrates several advantages over its counterparts: (1) *Larger data scale:* Our dataset stands as the largest multimodal dataset to date, containing 8.6 billion images, 1,696 billion text tokens, and 2.2 billion documents. It is 1.7 times larger in images and 12.5 times larger in texts compared to the previously largest multimodal dataset, LAION-5B (Schuhmann et al., 2022), while maintaining excellent data quality. (2) *Richer data diversity*: Drawing from a broader range of data sources, our dataset is more diverse than other image-text interleaved datasets. It in-

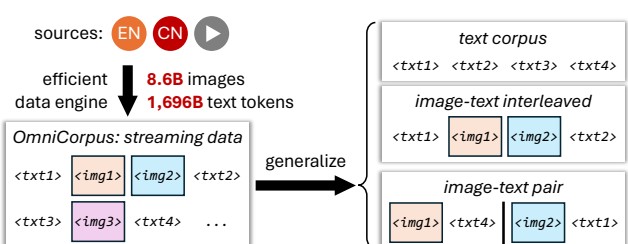

Figure 1: **Overview of our OmniCorpus dataset.** It comprises 8.6 billion images and 1,696 billion text tokens sourced from diverse origins. Additionally, our efficient data engine generalizes the data into various formats, such as text corpus, image-text interleaved, and image-text pairs.

cludes bilingual multimodal data in both Chinese and English, and encompasses text-centric and vision-centric documents extracted from common websites and video platforms. (3) *More flexible format*: The streaming data format of our dataset offers exceptional flexibility, allowing adaptation to various data structures, including pure text corpora, image-text pairs, and interleaved data formats.

We follow established practices (*e.g.*, LAION (Schuhmann et al., 2022), DataComp (Gadre et al., 2023), MMC4 (Zhu et al., 2024), and OBELICS (Laurençon et al., 2024a)) to responsibly handle privacy, safety, and data release. As with all web-crawled datasets, although it is impractical to obtain explicit consent from all content creators, we have spared no effort to comply with terms of use (see

Table 1: **Comparison with large-scale image-text pre-training datasets**. "#" denotes "the number of". "#Avg." denotes "#Images per sample | #Tokens per sample". The concept of "#Docs" applies only to interleaved image-text datasets and is not relevant to paired image-text datasets. The proposed OmniCorpus dataset features a significantly larger scale and a broader range of sources compared to previous image-text datasets.

| Dataset | #Images | #Tokens | #Docs | #Avg. | Language | Source |
|---|---|---|---|---|---|---|
| *Image-text Paired Datasets* | | | | | | |
| COYO-700M | 747M | 12.9B | – | 1 \| 17 | English | Common Crawl |
| LAION-5B | 5B | 135B | – | 1 \| 27 | multilingual | Common Crawl |
| *Image-text Interleaved Datasets* | | | | | | |
| KOSMOS-1 data | – | – | 71M | – \| – | English | Common Crawl |
| M3W (Flamingo) | 185M | – | 43M | 4.3 \| – | English | English Websites |
| Web Interleaved (MM1) | 1B | 500B | 500M | 2 \| 1K | English | English Websites |
| MMC4-Core | 29.9M | 2.4B | 7.3M | 4.1 \| 329 | English | Common Crawl |
| MMC4 | 585M | 43B | 103M | 5.7 \| 417 | English | Common Crawl |
| OBELICS | 353M | 115B | 141M | 2.5 \| 816 | English | Common Crawl |
| OmniCorpus-YT (ours) | 2.1B | 7.7B | 10M | 210 \| 770 | English | YouTube Videos (YT) |
| OmniCorpus-CW (ours) | 3.2B | 940B | 1196M | 2 \| 330 | Chinese | Chinese Websites (CW) |
| OmniCorpus-CC (ours) | 3.3B | 748B | 988M | 3.3 \| 757 | English | Common Crawl (CC) |
| **OmniCorpus (ours)** | **8.6B** | **1696B** | **2.2B** | **3.9 \| 574** | **Bilingual** | **CC, CW, YT** |

Appendix A.3). We have released all the processed documents with detailed attributes and curate higher-quality subsets filtered based on the attributes (see Appendix A.2). We hope that OmniCorpus will be a valuable resource for multimodal machine learning research.

In summary, our contributions are threefold:

(1) We introduce the OmniCorpus dataset, the largest open-source multimodal dataset to date, which pushes the boundaries of scale and diversity by encompassing 8.6 billion images interleaved with 1,696 text tokens from diverse sources, significantly surpassing previous datasets.

(2) We propose a comprehensive set of tools and algorithms, including a streaming data format that unifies multimodal data from various sources, an efficient and scalable data engine capable of processing large-scale data, and human feedback filters to ensure high-quality data.

(3) Through extensive experiments, we validate the quality and effectiveness of our dataset. We show that image-text interleaved data enhances few-shot capabilities and maintains the language abilities of multimodal models. Additionally, we also gained some new findings that differ from prior findings.

## 2 RELATED WORKS

### 2.1 IMAGE-TEXT DATASETS

As one of the three pillars of deep learning, datasets play a critical role in advancing deep learning models, especially in vision-language models (VLMs). Prior to the era of large-scale models, image-text datasets (Chen et al., 2015; Young et al., 2014; Goyal et al., 2017; Singh et al., 2019; Marino et al., 2019; Schwenk et al., 2022; Masry et al., 2022; Mishra et al., 2019; Wang et al., 2020; Clark & Gardner, 2018; Mathew et al., 2022) are primarily human-annotated and have limited data scale. For example, VQAv2 (Goyal et al., 2017) annotated each image with several question-answer pairs, while Visual Genome (Krishna et al., 2017) further provided region-level annotations. However, these datasets have limited data scales and fail to encompass diverse scenarios in the open world, hindering models' generalization ability. To achieve open-world capability, CLIP (Radford et al., 2021) and ALIGN (Jia et al., 2021) proposed training models using web-scale image-text pairs collected from the internet. Subsequent works (Schuhmann et al., 2021; 2022; Schuhman et al., 2022; Gadre et al., 2023; Byeon et al., 2022; Sharma et al., 2018; Changpinyo et al., 2021; Kalkowski et al., 2015; Thomee et al., 2016; Wang et al., 2024c;b; Peng et al., 2023) have also been introduced for open-source research. Among them, LAION-5B (Schuhmann et al., 2022) is the pioneering dataset offering billion-scale image-text pairs, whereas AS-1B (Wang et al., 2024c) is the first extensive dataset to provide region-level image-text pairs. By incorporating temporal context, InternVid (Wang et al., 2023c) and AuroraCap (Chai et al., 2024) enrich the diversity of visual content, further supporting the development of VLMs. However, these datasets contain limited world knowledge in each sample, affecting the performance of the underlying language model of VLMs. Recently,

Figure 2: **Overview of the data processing pipeline.** It contains five key stages: main body extraction, preliminary text filtering, document deduplication, image downloading & filtering, and detailed text filtering. Each stage efficiently reduces the dataset to retain only high-quality data.

a series of interleaved datasets (Zhu et al., 2024; Laurençon et al., 2024a) have been proposed to address these issues. Nonetheless, the data source and the languages involved in these datasets are limited. In this work, we propose the OmniCorpus, the first 10 billion-level image-text interleaved dataset comprising multiple data sources and languages.

## 2.2 VISION-LANGUAGE MODELS

Significant advancements have been made in the field of vision-language models (VLMs) in recent years. Previous methods (Bao et al., 2022; Wang et al., 2022) mainly focused on specific downstream tasks within predefined closed sets, while recent works have shifted towards understanding the open world. Models trained with contrastive learning-based methods (Radford et al., 2021; Jia et al., 2021; Fang et al., 2022; Chen et al., 2023b) are capable of recognizing and understanding open-world semantics through an image-text matching framework, although their lack of generative ability limits their applicability. In recent years, the advancement of large language models (LLMs) (Brown et al., 2020; Achiam et al., 2023; Touvron et al., 2023a) has led to the emergency of many LLM-based VLMs (Zhu et al., 2022; Li et al., 2023b; Zhu et al., 2023a; Wang et al., 2023b; Liu et al., 2023g; Li et al., 2023c). As one of the representative works, InternVL series model (Chen et al., 2024c; Gao et al., 2024; Chen et al., 2024b; Wang et al., 2024a) achieves performance comparable to GPT-4V (OpenAI, 2023). Additionally, models like Kosmos-2 (Peng et al., 2023) and ASMv2 (Wang et al., 2024b) enable LLMs to comprehend specific regions within images. Recently, a series of works (Sun et al., 2023c;a; Tian et al., 2024; Zhu et al., 2023b; Jin et al., 2023; Dong et al., 2023; Laurençon et al., 2024b) have explored the use of image-text interleaved data to enhance VLM capabilities. However, the training corpora for these models remain limited to English data from Common Crawl. The effectiveness of image-text interleaved data from other sources or languages is still unexplored. In this work, we provide more empirical insights into the use of interleaved data.

## 3 DATA ENGINE

### 3.1 OVERALL PIPELINE

Figure 2 illustrates the overall pipeline of our data engine, which consists of five key stages as follows:

**Main Body Extraction.** We extract primary content from each web document using an improved version of Trafilatura (Barbaresi, 2021), which can more accurately and efficiently extract main content and images while handling a broader range of languages (see Section 3.2). We enhance sections based on the HTML structure's density if the extracted content is insufficient. HTML documents without images are dropped in this stage. Some explicit advertisements or sidebars are excluded through HTML structure analysis and URL pattern matching for images. Then, we convert the HTML structure into the streaming data format, which is a unified data format applicable to different data sources. It preserve tags for individual elements, including <text>, <image>, , <header>, <detail>, <quote>, <video>, <audio>, <table>, and <list>. During this step, we remove 47% of documents.

**Preliminary Text Filtering.** Given the streaming data from the main body extraction, we perform preliminary text filtering by employing strategies from Gopher (Rae et al., 2021) and C4 (Raffel et al., 2020) to eliminate extremely low-quality content, such as documents with excessive numbers,

documents with texts that are too long or too short, documents containing explicit inaccurate content, and documents containing "lorem ipsum." Additionally, we introduce some heuristic rules to further filter the text, such as removing documents with too many continuous line breaks or documents where a single word's frequency is excessively high. During this step, we remove 80% documents from the remaining HTML documents.

**Document Deduplication with Text.** We remove duplicate documents by comparing their text content using minhash (Broder, 1997) values with a threshold of 0.8 and retaining the latest version. This step significantly reduces redundancy, discarding approximately 90% of duplicates.

**Image Downloading & Filtering.** In this step, we discard invalid images that were not successfully downloaded. Adhering to MMC4 (Zhu et al., 2024) guidelines, we filter out images with a height or width of fewer than 150 pixels and an aspect ratio greater than 2 or less than 0.5. We filtered out low-quality images using the LAION-Aesthetic Predictor (Schuhmann et al., 2022) with a threshold of 3.7 and the LAION-NSFW Detector (Schuhmann et al., 2022) with a threshold of 0.8. Additionally, we identify and remove images that appear more than 10 times across HTML documents by computing perceptual hash (phash) and difference hash (dhash) values.

**Detailed Text Filtering.** We use BERT (Devlin et al., 2018) classifiers of WanJuan-CC (Qiu et al., 2024) to score advertisement content, political content, toxic content, NSFW material, and document fluency. Using these models, we discard documents containing excessive ads, inappropriate content, or poor language quality. In addition, to further improve data quality, we use a human-feedback filtering strategy (see Section 3.3) to develop a multimodal filter suitable for English and non-English content.

In addition, we enhance the diversity of our dataset by creating storyboard datasets from various video sources. This includes extracting keyframes and transcribing audio content from YT-Temporal-1B (Zellers et al., 2022), HD-VILA-100M (Xue et al., 2022), HowTo100M (Miech et al., 2019), and InternVid (Wang et al., 2023c).

## 3.2 TWEAKINGS

To enhance the effectiveness and efficiency of our pipeline, we carefully refine the data pipeline from key aspects as follows:

**Pre-Deduplication.** The resources required for image downloading, filtering, and detailed text filtering are substantial, involving significant bandwidth, GPU resources, and human feedback. Given that the deduplication step filters out a large number of documents and images, we choose to perform deduplication in advance. This approach effectively reduces the number of images to be downloaded and the volume of documents requiring detailed text filtering. As a result, it saves approximately 86 PB seconds of bandwidth in downloading images, 4500 A100 GPU days in image filtering, and 130 GPU days along with 45 person-days in detailed text filtering.

**Improved Main Body Extraction.** Our extraction algorithm has been significantly improved compared to the vanilla Trafilatura (Barbaresi, 2021). In terms of accuracy, we have addressed the issue where Trafilatura would overlook the main content of an HTML document when extracting images, and enhanced its capability to handle Chinese, Japanese, and Arabic documents. Additionally, we have incorporated techniques to trim web noise regions based on HTML structure (such as clusters of lists and navigation bars) and style (targeting elements like advertisements, comments, JavaScript, and CSS). In terms of efficiency, we optimized the process based on HTML nodes and streamlined the processing pipeline by eliminating the fallback process in challenging cases. With these two improvements, we can not only extract more informative content from the main body but also double the speed of the extraction process.

**Improved Image Downloading.** We integrate efficient download task scheduling and resource allocation while employing Bloom filtering technology (Bloom, 1970) to deduplicate URLs of images that have been downloaded or are pending processing. This method effectively prevents redundant download requests, optimizing storage resources and bandwidth usage. Consequently, it provides robust technical support for the efficient collection and analysis of large-scale image data. Specifically, our approach reduces URL download requests from 30 billion to 9.65 billion and accelerates the downloading process by a factor of 1.5.

**Pipeline Parallelism.** Our pipeline runs in a modular parallel manner, offering several benefits. (1) The system will have greater fault tolerance since we can modify or improve each section of the pipeline independently. (2) Different parts of the pipeline require different types of resources, such as main body extraction runs on CPUs, image filtering runs on GPUs, and image downloading requires bandwidth, so a modular design is more reasonable. (3) by allocating resources based on throughput rather than evenly distributing them, we can significantly speed up the process. Compared to equal resource allocation, our parallel assembly line achieves a 1.39 times speed increase.

With all these improvements, the dataset processing pipeline can now scale up to thousands of CPUs, thousands of GPUs, and 3Gbps bandwidth, tripling its processing speed in that configuration. Further analysis of effectiveness is presented in Appendix C.1.

## 3.3 HUMAN-FEEDBACK FILTERING

Based on the pipeline introduced in Section 3.1, a significant portion of low-quality data has been removed. However, the remaining documents are still noisy. In this section, we introduce the human-feedback filtering method used to optimize the text filters, further improving the document quality. The optimized filter comprises nearly 30 filtering rules for English and 40 for Chinese. These filtering rules can be found in the Appendix.

To build these filtering rules, we first sample a subset of documents according to various criteria, including completeness, comprehensibility, fluency, relevance, and safety. After that, we manually design additional filtering rules to remove the low-quality documents from these sampled documents. These rules are then evaluated on a human-annotated evaluation set, and those achieving excellent performance are added to our filtering pipeline. The evaluation metric includes the miss rate and the false positive rate. By repeating the above process, we can iteratively optimize the quality and comprehensiveness of text filters based on human feedback.

## 3.4 STREAMING DATA FORMAT

We use a comprehensive and unified streaming data format to preserve rich and diverse information about the original data. Given an HTML document, we first split it into several chunks according to its layout, each formulated as image-text interleaved sequences $x = (x_1, x_2, ..., x_n)$, where $x_i$ can be a text sentence or an image. Then we concatenate these chunks in a top-to-bottom, left-to-right order to obtain a streaming interleaved sequence.

Based on this data format, the formulation of HTML documents, image-text pairs, and video sequences can be easily unified, which means that we can process these heterogeneous data from different sources in a unified manner. In addition to the content of the given data, other meta-annotations, including image aesthetic scores, image/text NSFW scores, political scores, toxic scores, unsafe scores, and text fluency, are also included in the streaming data. We hope that these meta-annotations can help researchers to better understand and utilize the dataset for various applications.

## 4 EXPLORING OMNICORPUS

**General Statistics.** As shown in Table 1, our OmniCorpus is currently the largest and the first open-source multilingual interleaved dataset. It surpasses the combined totals of all previous interleaved datasets (Huang et al., 2023; McKinzie et al., 2024; Zhu et al., 2024; Laurençon et al., 2024a). Figure 3 illustrates the joint distribution of text tokens and images in the interleaved sequences from OmniCorpus. See Appendix D.2 for more details.

**Diversity Analysis.** To measure and analyze the diversity of document content, we follow previous studies (Zhu et al., 2024; Laurençon et al., 2024a) and employ Latent Dirichlet Allocation (LDA) (Blei et al., 2003) to assess the topic diversity of the dataset. Figure 5 illustrates the significant differences in topics across documents from different sources, highlighting the importance of various sources in enhancing data diversity. The detailed topic modeling results are presented in Appendix D.3.

**Qualitative Assessment of Dataset Samples.** We randomly sample 200 documents from OmniCorpus-CC to evaluate their quality. There are 405 images in these documents. Among them, 88.4% are relevant to the documents, 8.0% contain watermarks, 4.0% contain logos, and 0.2%

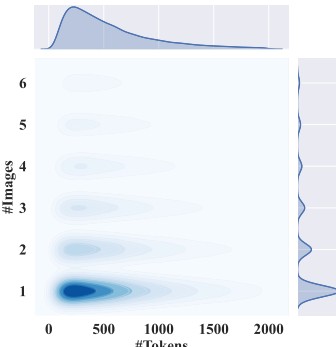

Figure 3: **Joint distribution of the image and token numbers per document.** We use kernel density estimate to get the distribution.

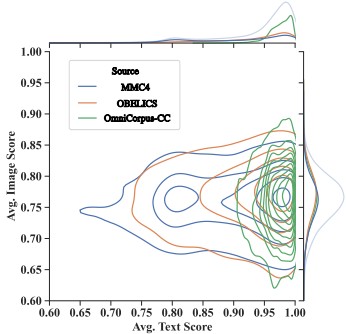

Figure 4: **Joint distribution of text and image score PDFs.** We visualized and compared the joint distribution of the PDFs of the Text Scores and Image Scores across each dataset.

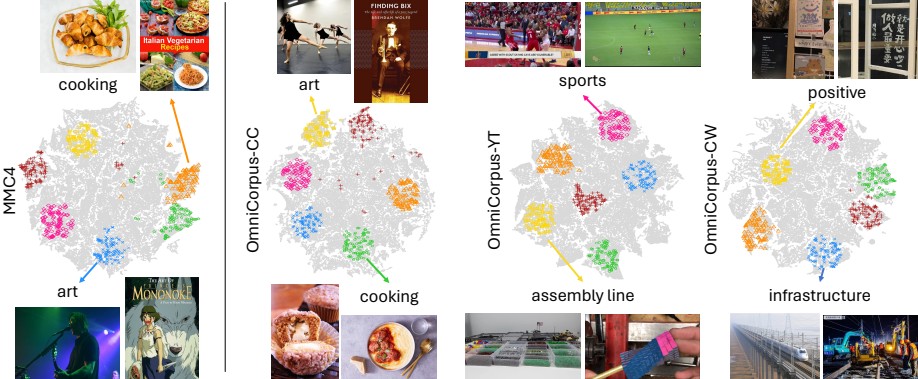

Figure 5: **Visualization of topic clusters and example images.** The four diagrams from left to right correspond to MMC4 (Zhu et al., 2024), OmniCorpus-CC, OmniCorpus-YT, and OmniCorpus-CW. The clusters are T-SNE (Van der Maaten & Hinton, 2008) projection of LDA (Blei et al., 2003) topic modeling results. We select 2 topics for each dataset and show two image examples for each topic.

are advertisements. Additionally, 86.4% of the documents feature photographic images, while 13.6% included graphic images such as cartoons. Furthermore, 32.1% of the images contain at least one written word, and 22.7% of the images contain structured text. No NSFW images were found.

**Quality Validation.** As illustrated in Figure 4, we present the joint distribution of text scores and image scores across each set of 1 million sampled documents. The image score is calculated as the average of the aesthetic score and the NSFW score. The text score is determined by averaging the advertisement content score, the NSFW content score, and the document fluency score. In terms of image scores, all datasets perform similarly. The OmniCorpus-CC exhibits superior text quality. Specifically, our OmniCorpus-CC has a lower proportion of low-quality text compared to other datasets, with the difference diminishing as test quality increases. This indicates a higher proportion of high-quality tests in OmniCorpus-CC.

## 5 EXPERIMENTS

In Section 5.2, we first run ablations on OmniCorpus and highlight key findings. In Section 5.3, we present results comparing MLLMs pre-trained on OmniCorpus with counterparts. We provide additional comparisons on instruction tuning in Appendix B.3 and analyses on effectiveness of data engine in Appendix C.

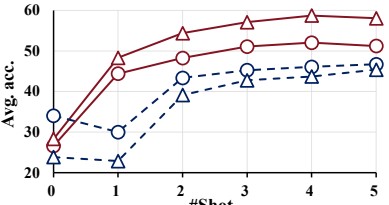

Figure 6: **Ablation on image position strategies.** Red solid: fully autoregressive. Blue dashed: cross-attention. Triangular: natural. Circular: retrieval-based.

Table 2: **Ablation on pre-training and SFT data types.** We report the zero/few-shot average accuracies of the four MLLM benchmarks and the text-only MMLU benchmark. The first row hosts the initialized model which has not been trained with vision-language data.

| Pre-training Data | SFT Data | Avg. MLLM acc. | | | | MMLU acc. | |
|---|---|---|---|---|---|---|---|
| | | 0 | 1 | 2 | 4 | 0 | 5 |
| - | - | - | - | - | - | 48.7 | 49.9 |
| Interleaved | - | 28.3 | 48.3 | 54.4 | 58.7 | 47.1 | 48.6 |
| - | Common | 76.3 | 71.7 | 72.6 | 73.1 | 50.3 | 50.5 |
| Interleaved | Common | 76.5 | 73.0 | 73.3 | 73.9 | 50.4 | 51.2 |
| Interleaved | Interleaved | 74.5 | 77.7 | 78.1 | 77.9 | 50.8 | 51.3 |

Table 3: **Pre-training ablation on curated subsets.** We report the zero/few-shot results on four MLLM benchmarks, including two VQA and two image captioning tasks. The first column shows the number of documents per subset, with 1M documents randomly sampled for training.

| Eval Set | OKVQA | | | | TextVQA | | | | COCO | | | | Flickr30k | | | | Avg. | | | |
|---|---|---|---|---|---|---|---|---|---|---|---|---|---|---|---|---|---|---|---|---|
| #Shot | 0 | 1 | 2 | 4 | 0 | 1 | 2 | 4 | 0 | 1 | 2 | 4 | 0 | 1 | 2 | 4 | 0 | 1 | 2 | 4 |
| 988M | 15.2 | 34.1 | 31.8 | 32.8 | 21.7 | 30.5 | 34.6 | 37.7 | 41.9 | 73.6 | 85.0 | 94.9 | 34.2 | 41.4 | 47.5 | 52.6 | 28.2 | 44.9 | 49.7 | 54.5 |
| 600M | **17.1** | 34.9 | 32.3 | 30.1 | **23.0** | 31.7 | 35.8 | 37.9 | 41.4 | 75.3 | 85.7 | 96.9 | 34.2 | 43.6 | 48.8 | **55.8** | **28.9** | 46.4 | 50.6 | 55.1 |
| 200M | 12.7 | **36.0** | 38.8 | 41.1 | 17.7 | 32.6 | **38.0** | **42.0** | **46.9** | **80.8** | **92.2** | **97.2** | **36.1** | 43.9 | 48.6 | 54.3 | 28.3 | **48.3** | **54.4** | **58.7** |
| 40M | 13.4 | 35.5 | 38.6 | **41.4** | 17.1 | 32.1 | 35.9 | 39.4 | 38.3 | 79.8 | 91.6 | 96.0 | 29.5 | **44.0** | 47.7 | 53.6 | 24.6 | 47.8 | 53.5 | 57.6 |
| 8M | 12.2 | 35.6 | 38.2 | 40.8 | 15.9 | 32.9 | 36.3 | 38.2 | 41.5 | 78.2 | 89.4 | 93.5 | 32.4 | 42.9 | **49.0** | 51.6 | 25.5 | 47.4 | 53.2 | 56.0 |
| 2.5M | 13.5 | 35.7 | **39.1** | 41.3 | 18.2 | **33.2** | 37.7 | 41.1 | 46.4 | 78.9 | 91.9 | 95.9 | 35.4 | 43.7 | 48.8 | 54.5 | 28.4 | 47.9 | 54.4 | 58.2 |

## 5.1 EXPERIMENTAL SETTINGS

**Baselines.** We construct our baseline models following LLaVA-1.5 (Liu et al., 2023d), which comprises a vision encoder, a multimodal projector, and an LLM. The input sequence to the LLM is a token sequence consisting of interleaved visual and textual tokens. The language modeling loss is used to train the model, which is only calculated on text tokens. Unless otherwise specified, we employ CLIP-ViT-L-336px (Radford et al., 2021) as the vision encoder and Vicuna-1.5-7B (Zheng et al., 2024) as the LLM.

**Evaluation.** We evaluate our models on VQA benchmarks (Goyal et al., 2017; Singh et al., 2019; Gurari et al., 2018; Marino et al., 2019) and image captioning benchmarks (Chen et al., 2015; Young et al., 2014). The accuracy score is used for VQA, while CIDEr (Vedantam et al., 2015) is used for image captioning. Following OpenFlamingo (Awadalla et al., 2023), we extend the benchmarks to few-shot settings to assess in-context learning. In Table 2-4, "Avg. MLLM acc." means the mean value of the four benchmark scores. In Table 2-5, "#Shot" means the number of in-context examples. See Appendix B.1 for more details.

## 5.2 MAIN FINDINGS

**Different image position strategies excel in different architectures.** Existing multimodal document datasets organize interleaved image and text sequences in two main ways. The MMC4 dataset (Zhu et al., 2024) employed a retrieval strategy, inserting images into text sequence based on CLIP similarities, while the OBELICS dataset (Laurençon et al., 2024a) maintained the natural layout of the source webpage. We conducted ablation studies on OmniCorpus-CC to evaluate both strategies using a fully autoregressive architecture like LLaVA-1.5 (Liu et al., 2023d) and a cross-attention architecture like Flamingo (Alayrac et al., 2022). As shown in Figure 6, the natural strategy performs better with the fully autoregressive architecture, whereas the retrieval-based strategy excels with the cross-attention architecture. This suggests that the cross-attention architecture benefits from optimal correlation between images and their surrounding paragraphs, while the fully autoregressive architecture prefers a natural arrangement that aligns with typical reading habits. Refer to Appendix B.5 for more details.

**Data filtering benefits MLLMs to some extent.** We further construct several curated subsets of approximately 600M, 200M, 40M, 8M, and 2.5M documents from OmniCorpus-CC. The curation details are introduced in Appendix B.4. To validate the benefits of data filtering, we trained baseline

Table 4: **Comparison with open-source interleaved image-text datasets.** We report the zero/few-shot results on four MLLM benchmarks. The best two results are highlighted with bold font.

| Eval Set | OKVQA | | | | TextVQA | | | | COCO | | | | Flickr30k | | | | Avg. | | | |
|---|---|---|---|---|---|---|---|---|---|---|---|---|---|---|---|---|---|---|---|---|
| #Shot | 0 | 1 | 2 | 4 | 0 | 1 | 2 | 4 | 0 | 1 | 2 | 4 | 0 | 1 | 2 | 4 | 0 | 1 | 2 | 4 |
| MMC4 | **15.1** | 29.0 | 24.0 | 23.2 | **21.2** | 27.6 | 30.3 | 33.8 | 45.7 | 70.9 | 82.1 | 88.4 | **36.3** | 32.5 | 39.0 | 43.8 | **29.6** | 40.0 | 43.9 | 47.3 |
| MMC4-Core | 13.5 | 29.5 | 27.1 | 26.8 | 20.5 | 27.1 | 32.1 | 35.6 | 41.0 | 72.1 | 84.6 | 90.3 | 34.3 | 37.5 | 41.1 | 45.6 | 27.3 | 41.5 | 46.2 | 49.6 |
| OBELICS | 13.9 | 35.0 | 36.8 | **40.2** | 17.9 | 30.3 | 35.7 | 40.7 | 50.7 | 74.7 | 91.3 | 97.1 | 42.7 | 41.4 | 47.5 | 54.7 | 31.3 | 45.3 | 52.9 | 58.2 |
| OmniCorpus-YT | **16.5** | **36.1** | **38.4** | 40.1 | **22.9** | **34.5** | **38.1** | **41.0** | 40.6 | 71.2 | 78.0 | 83.8 | 32.9 | 30.0 | 32.2 | 36.0 | 28.2 | 43.0 | 46.6 | 50.2 |
| OmniCorpus-CC | 12.7 | **36.0** | **38.8** | **41.1** | 17.7 | **32.6** | **38.0** | **42.0** | **46.9** | **80.8** | **92.2** | **97.2** | 36.1 | **43.9** | **48.6** | **54.3** | 28.3 | **48.3** | **54.4** | **58.7** |

Table 5: **Comparison with state-of-the-art MLLMs pre-trained with interleaved image-text data.** "*" indicates that the zero-shot evaluation follows Flamingo (Alayrac et al., 2022), which actually includes two text-only examples. The prompt for TextVQA (Singh et al., 2019) does not contain OCR tokens. To align with the evaluation setting of comparison models, we sample the in-context examples randomly.

| Model | Pre-training Data | #Shot | COCO | Flickr30k | OKVQA | TextVQA | VQAv2 | VizWiz |
|---|---|---|---|---|---|---|---|---|
| OpenFlamingo-9B | MMC4
LAION | 0* | 79.5 | 59.5 | 37.8 | 24.2 | 52.7 | 27.5 |
| | | 4 | 89.0 | 65.8 | 40.1 | 28.2 | 54.8 | 34.1 |
| | | 8 | 96.3 | 62.9 | 41.1 | 29.1 | 54.8 | 38.5 |
| IDEFICS-9B | OBELICS
Wikipedia
LAION, PMD | 0* | 46.0 | 27.3 | 38.4 | 25.9 | 50.9 | 35.5 |
| | | 4 | 93.0 | 59.7 | 45.4 | 27.6 | 55.4 | 36.9 |
| | | 8 | 97.0 | 61.9 | 47.7 | 27.5 | 56.4 | 40.4 |
| Emu-14B | LAION, LAION-COCO
MMC4, WebVid-10M
YT-Storyboard-1B | 0* | – | – | 42.8 | – | 52.9 | 34.4 |
| | | 4 | – | – | – | – | 58.4 | 41.3 |
| | | 8 | – | – | – | – | 59.0 | 43.9 |
| **Ours (7B)** | LAION
OmniCorpus-CC | 0* | **87.9** | **62.7** | **50.8** | **53.3** | **70.4** | **56.6** |
| | | 4 | **102.8** | **70.6** | **54.1** | **55.2** | **71.6** | **57.6** |
| | | 8 | **104.7** | **69.4** | **54.2** | **55.8** | **71.7** | **58.6** |

models using 1M documents randomly sampled from subsets, separately. As shown in Table 3, the model trained on the 200M subset outperforms those trained on larger subsets and performs similarly to the model trained on smaller subsets. This suggests that data filtering can improve data quality, but over-filtering may harm performance due to data homogenization.

**Image-text interleaved fine-tuning maintains in-context learning ability.** We pre-train the baseline architecture with 1M documents randomly sampled from OmniCorpus-CC and fine-tune it using the LLaVA-665K dataset (Liu et al., 2023d). We compare zero-shot and few-shot performance on four MLLM benchmarks, as well as a text-only benchmark (*i.e.*, MMLU (Hendrycks et al., 2020)), as shown in Table 2. The image-text interleaved pre-trained model shows a stepwise improvement with more in-context examples. After fine-tuning with high-quality conversation samples, there are overall enhancements for the average performance on four MLLM benchmarks, but the positive correlation with the example number is no longer maintained. Additionally, we replace the caption and VQA samples in the SFT data with few-shot samples whose format is aligned with the evaluation, yielding significantly improved few-shot performance. Despite the slight decline in zero-shot performance, the best few-shot average score shows considerable improvement compared to the baseline. Therefore, including image-text interleaved samples in SFT data is still essential. Furthermore, due to the absence of text-only instruction following samples in this setting, the model's language capability decreased. However, the high-quality data used in SFT significantly improved the language ability, effectively mitigating the disadvantages introduced during the pre-training phase.

**OmniCorpus-YT boosts VQA performance while degrading captioning ability.** The previous studies have merely incorporated storyboard samples into a pre-training data mixture without thoroughly investigating the specific impact. Our goal is to pre-train an MLLM exclusively using documents collected from video and evaluate it on image-text benchmarks. We randomly selected 1M samples from OmniCorpus-YT. For each sample of video frames with text, we uniformly extracted six frames as images for the document and removed the remaining frames, constructing an image-text interleaved document. As shown in Table 4, the model trained on sampled OmniCorpus-YT achieves the best VQA capabilities, but its captioning scores are the lowest. The results demonstrate the feasibility of extracting image-text interleaved documents from video resources.

**OmniCorpus-CW improves the Chinese ability.** We pre-train on 1M Chinese documents randomly sampled from OmniCorpus-CW and fine-tune with LLaVA-665K data (Liu et al., 2023d). We find that the scores improve from 59.8 to 62.5 (+2.7) for MMBench-CN (Liu et al., 2023f) and from 23.6 to 24.9 (+1.3) for CMMMU (Zhang et al., 2024), demonstrating the effectiveness of our OmniCorpus-CW data.

## 5.3 COMPARISON EXPERIMENTS

The OmniCorpus achieves a large data scale while ensuring superior data quality. To demonstrate this, we conduct comparison experiments on 1M documents randomly sampled from MMC4, MMC4-Core (Zhu et al., 2024), OBELICS (Laurençon et al., 2024a), and OmniCorpus-CC, respectively. The MLLM architectures and pre-training settings are kept consistent across all experiments. As is shown in Table 4, the OmniCorpus-CC exhibits optimal few-shot performance and near-optimal zero-shot performance. OmniCorpus-CC improves the capacity of in-context learning, which is widely acknowledged as a key advantage of pre-training with image-text interleaved data. Additionally, the larger scale of our dataset makes it particularly suitable for extensive multimodal pre-training.

To demonstrate the potential of the OmniCorpus for large-scale MLLMs pre-training, we design a recipe for training a competitive 7B baseline foundation model with our dataset. We replace the LLM with InternLM2-7B (Cai et al., 2024). Additionally, we collect a large-scale data mixture, including image-text interleaved data (OmniCorpus-CC), paired image-text data (LAION (Schuhmann et al., 2022)), and text-only data. We compare our model with OpenFlamingo (Awadalla et al., 2023) mainly pre-trained with MMC4 (Zhu et al., 2024) and IDEFICS mainly pre-trained with OBELICS (Laurençon et al., 2024a). We follow them to add two evaluation sets, VQAv2 (Goyal et al., 2017) and VizWiz (Gurari et al., 2018), for evaluating the pre-trained models. The evaluation setting is aligned with the OpenFlamingo (Awadalla et al., 2023). The comparison performance is presented in Table 5. We can see that our 7B model is superior to the larger 9B OpenFlamingo and IDEFICS in most cases. Especially for VQAv2 and TextVQA, our model achieves a cliff lead.

## 6 CONCLUSION

In this work, we introduce the OmniCorpus dataset, the largest multimodal dataset to date. This dataset contains 8.6 billion images, 1,696 billion text tokens, and 2.2 billion documents, which are collected from three data sources: Common Crawl, Chinese websites, and video platforms. We elaborate on the data engine used to construct this dataset and carefully analyze its diversity and quality. Experimental results demonstrate the effectiveness of our OmniCorpus. We also provide some new insights according to these experiments.

## ACKNOWLEDGMENTS

This project was supported by the National Key R&D Program of China (No. 2022ZD0161301, 2022ZD0160101), the National Natural Science Foundation of China (No. 62376134). Zhe Chen was supported by the Youth PhD Student Research Project under the National Natural Science Foundation (No. 623B2050).

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

# A    DATASET INFORMATION

## A.1    DATASHEET FOR OMNICORPUS DATASET

### A.1.1    MOTIVATION

Q1 **For what purpose was the dataset created?** Was there a specific task in mind? Was there a specific gap that needed to be filled? Please provide a description.

- OmniCorpus was created to address the limitations of existing image-text interleaved datasets, specifically their scale and diversity. The dataset contains 10 billion-level image-text pairs, with the goal of enhancing multimodal large language models (MLLMs). Unlike previous datasets that often focus on English and text-centric sources, OmniCorpus includes a broad range of data from both English and non-English websites as well as video-centric platforms, providing a more diverse and comprehensive resource for training MLLMs. The dataset's flexibility in data formats (pure text corpus, image-text pairs, and interleaved data) aims to support various research applications in multimodal learning.

Q2 **Who created the dataset (e.g., which team, research group) and on behalf of which entity (e.g., company, institution, organization)?**

- The OmniCorpus dataset was created by OpenGVLab, the general vision team of Shanghai AI Laboratory.

Q3 **Who funded the creation of the dataset?** If there is an associated grant, please provide the name of the granter and the grant name and number.

- This project was supported by the National Key R&D Program of China (No. 2022ZD0161301, 2022ZD0160101), the National Natural Science Foundation of China (No. 62376134). Zhe Chen was supported by the Youth PhD Student Research Project under the National Natural Science Foundation (No. 623B2050).

Q4 **Any other comments?**

- No.

### A.1.2    COMPOSITION

Q5 **What do the instances that comprise the dataset represent (e.g., documents, photos, people, countries)?** *Are there multiple types of instances (e.g., movies, users, and ratings; people and interactions between them; nodes and edges)? Please provide a description.*

- Each instance in OmniCorpus represents an image-text interleaved document. These instances include a variety of image types and corresponding textual descriptions. The dataset is diverse, encompassing images and text from English and non-English websites, as well as video platforms. The data is structured in a streaming format that allows for various configurations, such as pure text corpora, image-text pairs, and interleaved sequences.

Q6 **How many instances are there in total (of each type, if appropriate)?**

- OmniCorpus consists of 8.6 billion images, 1,696 billion text tokens, and 2.2 billion documents. The dataset is significantly larger and more diverse compared to previous multimodal datasets.

Q7 **Does the dataset contain all possible instances or is it a sample (not necessarily random) of instances from a larger set?** *If the dataset is a sample, then what is the larger set? Is the sample representative of the larger set (e.g., geographic coverage)? If so, please describe how this representativeness was validated/verified. If it is not representative of the larger set, please describe why not (e.g., to cover a more diverse range of instances, because instances were withheld or unavailable).*

- OmniCorpus is a sample from Common Crawl (Common Crawl, 2007), Chinese websites, YT-Temporal-1B (Zellers et al., 2022), HD-VILA-100M (Xue et al., 2022), HowTo100M (Miech et al., 2019), and InternVid (Wang et al., 2023c). The data was

filtered and processed to maintain high quality and relevance, though it may not capture every possible instance from the larger set of internet data.

Q8 **What data does each instance consist of?** *"Raw" data (e.g., unprocessed text or images) or features? In either case, please provide a description.*

- Each instance consists of an interleaved sequence of images and text. The data includes raw image URLs, associated text, and metadata such as image dimensions, language, and various quality scores. The text can be captions, descriptions, or other types of annotations related to the images.

Q9 **Is there a label or target associated with each instance?** *If so, please provide a description.*

- No, OmniCorpus does not provide specific labels or targets for each instance. The dataset is designed to be flexible and can be used for various tasks such as image recognition, captioning, and visual question answering, depending on the researcher's needs.

Q10 **Is any information missing from individual instances?** *If so, please provide a description, explaining why this information is missing (e.g., because it was unavailable). This does not include intentionally removed information, but might include, e.g., redacted text.*

- No.

Q11 **Are relationships between individual instances made explicit (e.g., users' movie ratings, social network links)?** *If so, please describe how these relationships are made explicit.*

- No.

Q12 **Are there recommended data splits (e.g., training, development/validation, testing)?** *If so, please provide a description of these splits, explaining the rationale behind them.*

- No.

Q13 **Are there any errors, sources of noise, or redundancies in the dataset?** *If so, please provide a description.*

- OmniCorpus is generated through a data engine and may contain some noise or errors. However, efforts were made to filter and clean the data, including human feedback and filtering processes. Although the noise has been significantly reduced, we encourage the users to further filter the data based on their requirements.

Q14 **Is the dataset self-contained, or does it link to or otherwise rely on external resources (e.g., websites, tweets, other datasets)?** *If it links to or relies on external resources, a) are there guarantees that they will exist, and remain constant, over time; b) are there official archival versions of the complete dataset (i.e., including the external resources as they existed at the time the dataset was created); c) are there any restrictions (e.g., licenses, fees) associated with any of the external resources that might apply to a future user? Please provide descriptions of all external resources and any restrictions associated with them, as well as links or other access points, as appropriate.*

- The dataset relies on URLs to images hosted on the web. While the data was collected to be as stable as possible, there are no guarantees that all external resources will remain available indefinitely. The dataset includes URLs and annotations, but not the media content itself. Users must respect the original sources' licenses and restrictions when accessing the data.

Q15 **Does the dataset contain data that might be considered confidential (e.g., data that is protected by legal privilege or by doctor-patient confidentiality, data that includes the content of individuals' non-public communications)?** *If so, please provide a description.*

- No.

Q16 **Does the dataset contain data that, if viewed directly, might be offensive, insulting, threatening, or might otherwise cause anxiety?** *If so, please describe why.*

- The dataset includes images and text from various internet sources, and despite filtering efforts, it may still contain content that some users might find offensive or distressing. Hence, a subset with higher scrutiny and manual verification is available to minimize exposure to such content. Although the content has been significantly reduced, we encourage the users to further filter the data based on their requirements.

Q17 **Does the dataset relate to people?** *If not, you may skip the remaining questions in this section.*

- People may appear in images or be mentioned in text, but they are not the primary focus of the dataset.

Q18 **Does the dataset identify any subpopulations (e.g., by age, gender)?**

- The dataset does not explicitly identify subpopulations. Any such information would be incidental and not a primary attribute of the dataset.

Q19 **Is it possible to identify individuals (i.e., one or more natural persons), either directly or indirectly (i.e., in combination with other data) from the dataset?** *If so, please describe how.*

- Yes, the dataset it possible to identify individuals because it comes from the internet.

Q20 **Does the dataset contain data that might be considered sensitive in any way (e.g., data that reveals racial or ethnic origins, sexual orientations, religious beliefs, political opinions or union memberships, or locations; financial or health data; biometric or genetic data; forms of government identification, such as social security numbers; criminal history)?** *If so, please provide a description.*

- Yes, the dataset includes images and text from various internet sources, and despite filtering efforts, it may still contain content that some users might find sensitive. Hence, a subset with higher scrutiny and manual verification is available to minimize exposure to such content. Although the content has been significantly reduced, we encourage the users to further filter the data based on their requirements.

Q21 **Any other comments?**

- No.

### A.1.3 COLLECTION PROCESS

Q22 **How was the data associated with each instance acquired?** *Was the data directly observable (e.g., raw text, movie ratings), reported by subjects (e.g., survey responses), or indirectly inferred/derived from other data (e.g., part-of-speech tags, model-based guesses for age or language)? If data was reported by subjects or indirectly inferred/derived from other data, was the data validated/verified? If so, please describe how.*

- The OmniCorpus dataset is directly observable, from Common Crawl (Common Crawl, 2007), chinese websites, YT-Temporal-1B (Zellers et al., 2022), HD-VILA-100M (Xue et al., 2022), HowTo100M (Miech et al., 2019), and InternVid (Wang et al., 2023c).

Q23 **What mechanisms or procedures were used to collect the data (e.g., hardware apparatus or sensor, manual human curation, software program, software API)?** *How were these mechanisms or procedures validated?*

- We ran the data engine in Python, over 128 8-A100(80G) GPU machine, 30,000 CPU machine and 3Gbps bandwidth.

Q24 **If the dataset is a sample from a larger set, what was the sampling strategy (e.g., deterministic, probabilistic with specific sampling probabilities)?**

- The OmniCorpus dataset is created based on the Common Crawl (Common Crawl, 2007) and YT-Temporal-1B (Zellers et al., 2022), HD-VILA-100M (Xue et al., 2022), HowTo100M (Miech et al., 2019), and InternVid (Wang et al., 2023c).

Q25 **Who was involved in the data collection process (e.g., students, crowdworkers, contractors) and how were they compensated (e.g., how much were crowdworkers paid)?**

- Almost all researchers and developers in this project have been involved in the author list. The work was done when the students including Qingyun Li, Zhe Chen, Weiyun Wang, Guanzhou Chen, Zhangwei Gao, Erfei Cui, Changyao Tian were research interns at Shanghai AI Laboratory. No crowdworkers were used in the curation of the dataset. The researchers and developers made academic contributions without additional payments.

Q26 **Over what timeframe was the data collected? Does this timeframe match the creation timeframe of the data associated with the instances (e.g., recent crawl of old news articles)?** *If not, please describe the timeframe in which the data associated with the instances was created.*

- The data for the OmniCorpus dataset was collected over a timeframe that encompasses multiple years, as it includes a vast and diverse range of sources such as Common Crawl, Chinese websites, and YouTube. This comprehensive collection effort aims to cover a wide spectrum of content types and contexts. The timeframe of the data collection does not necessarily match the creation timeframe of the data associated with the instances. For instance, the dataset includes recent crawls of older news articles and video frames extracted from previously published videos. This approach ensures the inclusion of both contemporary and historical content, thus providing a rich and varied dataset for training multimodal models.

Q27 **Were any ethical review processes conducted (e.g., by an institutional review board)?** *If so, please provide a description of these review processes, including the outcomes, as well as a link or other access point to any supporting documentation.*

- The project has been reviewed by the Ethics Review Committee of ICLR 2025. This paper was evaluated by at least two members of the Ethics Review Committee. Additionally, the reviewers and area chair have also fulfilled their responsibilities of conducting ethical reviews.

Q28 **Does the dataset relate to people?** *If not, you may skip the remaining questions in this section.*

- People might be present in the images and descriptions, although they are not the sole emphasis of the dataset.

Q29 **Did you collect the data from the individuals in question directly, or obtain it via third parties or other sources (e.g., websites)?**

- The data for OmniCorpus was obtained from third-party sources, including Common Crawl, Chinese websites, and YouTube, rather than collected directly from individuals.

Q30 **Were the individuals in question notified about the data collection?** *If so, please describe (or show with screenshots or other information) how notice was provided, and provide a link or other access point to, or otherwise reproduce, the exact language of the notification itself.*

- Individuals were not notified about the data collection. Our dataset is built upon Common Crawl (Common Crawl, 2007), chinese websites, YT-Temporal-1B (Zellers et al., 2022), HD-VILA-100M (Xue et al., 2022), HowTo100M (Miech et al., 2019), and InternVid (Wang et al., 2023c), which only contains information that is publicly available on the Internet. The publishers of these information are usually aware that it will be made public to the world, but they may not be aware that it will be collected in this way.

Q31 **Did the individuals in question consent to the collection and use of their data?** *If so, please describe (or show with screenshots or other information) how consent was requested and provided, and provide a link or other access point to, or otherwise reproduce, the exact language to which the individuals consented.*

- No. See Q30 for more details.

Q32 **If consent was obtained, were the consenting individuals provided with a mechanism to revoke their consent in the future or for certain uses?** *If so, please provide a description, as well as a link or other access point to the mechanism (if appropriate).*

- Users can contact the research team of the OmniCorpus for image(s) removal. Besides, users can contact us to remove any document in our proposed OmniCorpus.

Q33 **Has an analysis of the potential impact of the dataset and its use on data subjects (e.g., a data protection impact analysis) been conducted?** *If so, please provide a description of this analysis, including the outcomes, as well as a link or other access point to any supporting documentation.*

- No.

Q34 **Any other comments?**

- No.

### A.1.4 PREPROCESSING, CLEANING, AND/OR LABELING

Q35 **Was any preprocessing/cleaning/labeling of the data done (e.g., discretization or bucketing, tokenization, part-of-speech tagging, SIFT feature extraction, removal of instances, processing of missing values)?** *If so, please provide a description. If not, you may skip the remainder of the questions in this section.*

- Yes. The preprocessing involves several steps: main body extraction using an improved version of Trafilatura (Barbaresi, 2021), preliminary text filtering employing strategies from Gopher (Rae et al., 2021) and C4 (Raffel et al., 2020), document deduplication using minihash values, image downloading and filtering according to MMC4 (Zhu et al., 2024) guidelines and LAION-5B (Schuhmann et al., 2022), detailed text filtering based on BERT (Devlin et al., 2018) models, and human-feedback filtering to enhance data quality.

Q36 **Was the "raw" data saved in addition to the preprocessed/cleaned/labeled data (e.g., to support unanticipated future uses)?** *If so, please provide a link or other access point to the "raw" data.*

- No.

Q37 **Is the software used to preprocess/clean/label the instances available?** *If so, please provide a link or other access point.*

- Yes, all code has been open-sourced in our GitHub repository. See Section A.2 for more details.

Q38 **Any other comments?**

- No.

### A.1.5 USES

Q39 **Has the dataset been used for any tasks already?** *If so, please provide a description.*

- Yes, the OmniCorpus dataset has been used for training multimodal large language models (MLLMs), specifically demonstrating its effectiveness in vision-language tasks such as image captioning and visual question answering.

Q40 **Is there a repository that links to any or all papers or systems that use the dataset?** *If so, please provide a link or other access point.*

- No.

Q41 **What (other) tasks could the dataset be used for?**

- Long text-image retrieval: We provide image-text similarities calculated with CLIP, which can convert documents into an image-text retrieval dataset with longer text. A retrieval model pre-trained on such data can retrieve images based on longer text or textual documents, which can be used for multimodal retrieval-augmented generation, converting purely textual documents into multimodal documents, etc.
- Source for further dataset research: Our dataset is large-scale and can serve as a source for research on data curation strategies. We provide many useful attributes as metadata for each document, which can enrich the filtering strategy and reduce costs.

Q42 **Is there anything about the composition of the dataset or the way it was collected and preprocessed/cleaned/labeled that might impact future uses?** *For example, is there anything that a future user might need to know to avoid uses that could result in unfair treatment of individuals or groups (e.g., stereotyping, quality of service issues) or other undesirable harms (e.g., financial harms, legal risks) If so, please provide a description. Is there anything a future user could do to mitigate these undesirable harms?*

- Yes, the dataset includes data from diverse sources including non-English websites and video platforms, which enhances its diversity. However, the dataset also includes data from the internet which may contain biases or low-quality content. Measures have been taken to filter out low-quality and irrelevant content through human-feedback text filters.

Q43 **Are there tasks for which the dataset should not be used?** *If so, please provide a description.*

- As with all common practices for web-crawled image-text interleaved datasets, the OmniCorpus should not be used in any project that involves sensitive content or harmful outcomes. This includes but is not limited to applications in political manipulation, hate speech generation, misinformation propagation, or any tasks that may perpetuate harmful stereotypes or biases.

Q44 **Any other comments?**

- The legal statements and terms of use related to OmniCorpus are in Section E.

### A.1.6 DISTRIBUTION

Q45 **Will the dataset be distributed to third parties outside of the entity (e.g., company, institution, organization) on behalf of which the dataset was created?** *If so, please provide a description.*

- Yes, the dataset has be open-sourced on HuggingFace platform and OpenDataLab platform. See Section A.2 for more details.

Q46 **How will the dataset be distributed (e.g., tarball on website, API, GitHub)?** *Does the dataset have a digital object identifier (DOI)?*

- See Q45 for more details.

Q47 **When will the dataset be distributed?**

- The dataset has been distributed.

Q48 **Will the dataset be distributed under a copyright or other intellectual property (IP) license, and/or under applicable terms of use (ToU)?** *If so, please describe this license and/or ToU, and provide a link or other access point to, or otherwise reproduce, any relevant licensing terms or ToU, as well as any fees associated with these restrictions.*

- We detail the license, legal statements, and terms of use in Section E.

Q49 **Have any third parties imposed IP-based or other restrictions on the data associated with the instances?** *If so, please describe these restrictions, and provide a link or other access point to, or otherwise reproduce, any relevant licensing terms, as well as any fees associated with these restrictions.*

- OmniCorpus owns the metadata and release as CC BY 4.0 license.
- We do not own the copyright of the images.

Q50 **Do any export controls or other regulatory restrictions apply to the dataset or to individual instances?** *If so, please describe these restrictions, and provide a link or other access point to, or otherwise reproduce, any supporting documentation.*

- No.

Q51 **Any other comments?**

- No.

### A.1.7 MAINTENANCE

Q52 **Who will be supporting/hosting/maintaining the dataset?**

- OpenGVLab of Shanghai AI Laboratory will maintain the dataset.

Q53 **How can the owner/curator/manager of the dataset be contacted (e.g., email address)?**

- We recommend to contact us on the GitHub Issues[1].

Q54 **Is there an erratum?** *If so, please provide a link or other access point.*

- No.

Q55 **Will the dataset be updated (e.g., to correct labeling errors, add new instances, delete instances)?** *If so, please describe how often, by whom, and how updates will be communicated to users (e.g., mailing list, GitHub)?*

- No. However, specific samples can be removed on request.

Q56 **If the dataset relates to people, are there applicable limits on the retention of the data associated with the instances (e.g., were individuals in question told that their data would be retained for a fixed period of time and then deleted)?** *If so, please describe these limits and explain how they will be enforced.*

- People may contact us to add specific samples to a blacklist.

Q57 **Will older versions of the dataset continue to be supported/hosted/maintained?** *If so, please describe how. If not, please describe how its obsolescence will be communicated to users.*

- We will only support and maintain the latest version at all times, and a new version release of OmniCorpus will automatically deprecate its previous version.

Q58 **If others want to extend/augment/build on/contribute to the dataset, is there a mechanism for them to do so?** *If so, please provide a description. Will these contributions be validated/verified? If so, please describe how. If not, why not? Is there a process for communicating/distributing these contributions to other users? If so, please provide a description.*

- We welcome any contributions to OmniCorpus, and we will announce updates regarding dataset extensions on GitHub. However, contributors must demonstrate the quality and harmlessness of the extended data annotations; otherwise, we will not accept these extensions.

Q59 **Any other comments?**

- No.

## A.2 RELEASE AND MAINTAINING

We follow established practices of dataset research (*e.g.*, LAION (Schuhmann et al., 2022), DataComp (Gadre et al., 2023), MMC4 (Zhu et al., 2024), and OBELICS (Laurençon et al., 2024a)) to release our work. The navigation for all public resources of this project is available in the GitHub repository[2].

We have uploaded all processed documents to public data hosting platforms. Specifically, The OmniCorpus-CC[3] and the OmniCorpus-YT[4] are hosted on the HuggingFace platform. The OmniCorpus-CW[5] is hosted on the OpenDataLab platform. To reduce the cost of further processing for users, the meta-annotations (introduced in Section 3.4) contain many useful attributes, including document attributes (fluency, non-advertisement, pornography, politics, and toxicity probabilities), image attributes (aesthetic and punsafe probabilities, width and height, aspect ratio, file size, and repetition rates), and image-text similarities.

We also upload higher-quality subsets curated with the attributes, such as OmniCorpus-CC-210M[6] which is filtered from OmniCorpus-CC.

In additional to releasing data, we also consider uphold the transparency in data collection and the reproducibility of model results. The code for interleaved image-text pre-training with OmniCorpus,

---

[1]https://github.com/OpenGVLab/OmniCorpus/issues

[2]https://github.com/OpenGVLab/OmniCorpus

[3]https://huggingface.co/datasets/OpenGVLab/OmniCorpus-CC

[4]https://huggingface.co/datasets/OpenGVLab/OmniCorpus-YT

[5]https://openxlab.org.cn/datasets/Li-Qingyun/OmniCorpus-CW

[6]https://huggingface.co/datasets/OpenGVLab/OmniCorpus-CC-210M

along with scripts for few-shot evaluation, is provided in the GitHub repository. The developed human-feedback filtering functions and enhanced mainbody extraction tools are also available.

We are open to further refining our approach based on community feedback to maintain high ethical standards in the creation and distribution of OmniCorpus. We hope that OmniCorpus will be a valuable open resource for multimodal machine learning research.

### A.3 Ethical Discussion

During the collection and release of the OmniCorpus dataset, we place great importance on ethical considerations. In addition to following the established corpora (*e.g.*, MMC4 (Zhu et al., 2024) and OBELICS (Laurençon et al., 2024a)), we make additional efforts to uphold high ethical standards. We are open to further refining our approach while maintaining open-source resources based on community feedback.

We make substantial efforts to respect privacy by removing infringing content, including personal identifiers, phone numbers, bank accounts, emails, social media accounts, and content where opt-out signals are present. As all corpora sourced from the web (*e.g.*, LAION (Schuhmann et al., 2022) and OBELICS (Laurençon et al., 2024a)), it is impractical to obtain explicit consent from all content creators. The approach, while not exhaustive, reflects a commitment to respecting individual privacy and consent as much as possible.

To mitigate the inclusion of undesirable content, a rigorous filtering process was implemented. We filter out pornographic, fabricated, biases and gambling content as well as other potentially harmful material. We also exclude unreliable website domains that are more likely to contain inappropriate content. (For example, we exclude all content from disneylies.com, which claims that "All information on this site is false.".) Despite these efforts, the nature of web-crawled data means some inappropriate content might still be present. Continuous monitoring and updating of the filtering mechanisms are necessary to improve the dataset's quality and safety.

By addressing these ethical considerations, the OmniCorpus project strives to adhere to high standards for responsible data handling and usage in the realm of multimodal machine learning research.

## B Supplementary Experiment Details

### B.1 Evaluation Details

We evaluate the pre-trained models on four VQA benchmarks (including OKVQA (Marino et al., 2019), TextVQA (Singh et al., 2019), VQAv2 (Goyal et al., 2017), and VizWiz (Gurari et al., 2018)) and two image captioning benchmarks (including COCO Caption (Chen et al., 2015) and Flickr30K Caption (Young et al., 2014)). Since the baseline models in ablation experiments are based on LLaVA-1.5 (Liu et al., 2023d), we support RICES-based few-shot prompting (Yang et al., 2022) for the open-source evaluation tools of LLaVA-1.5, which do not post-process the response and use OCR tokens for TextVQA. When comparing with state-of-the-art MLLMs pre-trained with image-text interleaved data (in Table 5), we adapt our model to the open-source evaluation tools of OpenFlamingo (Awadalla et al., 2023), which sample few-shot examples randomly. For both settings, we provide few-shot examples in the chatting history of multi-round conversations. The formats of few-shot prompting for VQA and image captioning are provided in Table 6.

### B.2 Training Details

We build the baseline models based on the LLaVA-1.5 (Liu et al., 2023d). The models in ablation studies employ CLIP-ViT-L-336px (Radford et al., 2021) and Vicuna-1.5-7B (Zheng et al., 2024) as the vision encoder and the LLM, respectively. For the final model in Table 5, we replace them with InternViT-300M-448px (Chen et al., 2023b) and InternLM2-7B (Team, 2023). Additionally, we employ a two-layer MLP pre-aligned with captioning data as introduced in LLaVA-1.5. During the pre-training, we freeze the vision encoder and update the parameters of the MLP projector and the LLM. We train the models with 1 million image-text interleaved documents on 16 80GB A100 GPUs for about one day.

Table 6: **The formats of few-shot prompting for VQA and image captioning.** The demonstrated template is from Vicuna Chiang et al. (2023). Only one-shot situations are illustrated here; in practice, the number of turns varies based on the number of shots. $\mathbf{X}_{\texttt{system-message}}$ indicates the system message. The rest $\mathbf{V}$, $\mathbf{X}$, and $\mathbf{Y}$ represent the tokens for the image, prompt, and response for an example or a test sample, respectively. `<STOP>` represents stop indicators. The green tokens are the expected responses.

---

**VQA Prompt:**

$\mathbf{X}_{\texttt{system-message}}$ `<STOP>`

`Human`: $\mathbf{V}^1_{\texttt{shot}}$ $\mathbf{X}^1_{\texttt{shot}}$ `Answer the question using a single word or phrase.` `<STOP>`

`Assistant`: $\mathbf{Y}^1_{\texttt{shot}}$ `<STOP>`

`...`

`Human`: $\mathbf{V}_{\texttt{test}}$ $\mathbf{X}_{\texttt{test}}$ `Answer the question using a single word or phrase.` `<STOP>`

`Assistant:` $\mathbf{Y}_{\texttt{response}}$ `<STOP>`

---

**Image Captioning Prompt:**

$\mathbf{X}_{\texttt{system-message}}$ `<STOP>`

`Human`: $\mathbf{V}^1_{\texttt{shot}}$ `Provide a one-sentence caption for the provided image.` `<STOP>`

`Assistant`: $\mathbf{Y}^1_{\texttt{shot}}$ `<STOP>`

`...`

`Human`: $\mathbf{V}_{\texttt{test}}$ `Provide a one-sentence caption for the provided image.` `<STOP>`

`Assistant:` $\mathbf{Y}_{\texttt{response}}$ `<STOP>`

---

Table 7: **Results on 12 general visual-language benchmarks**. Benchmark names are abbreviated due to space limits. VQA-v2 (Goyal et al., 2017); GQA (Hudson & Manning, 2019); VizWiz (Gurari et al., 2018); SQA$^\mathrm{I}$: ScienceQA-IMG (Lu et al., 2022a); VQA$^\mathrm{T}$: TextVQA (Singh et al., 2019); POPE (Li et al., 2023d); MME (Fu et al., 2023); MMB: MMBench (Liu et al., 2023f); MMB$^\mathrm{CN}$: MMBench-Chinese (Liu et al., 2023f); SEED: SEED-Bench (Li et al., 2023a); LLaVA$^\mathrm{W}$: LLaVA-Bench (In-the-Wild) (Liu et al., 2023e); MM-Vet (Yu et al., 2023b). *The training images of the datasets are observed during training. The best performances are marked **bold**.

| Model | VQA$^{\mathrm{v2}}$ | GQA | VizWiz | SQA$^\mathrm{I}$ | VQA$^\mathrm{T}$ | POPE | MME | MMB | MMB$^\mathrm{CN}$ | SEED | LLaVA$^\mathrm{W}$ | MM-Vet |
|---|---|---|---|---|---|---|---|---|---|---|---|---|
| BLIP-2 | 41.0 | 41.0 | 19.6 | 61.0 | 42.5 | 85.3 | 1293.8 | — | — | 46.4 | 38.1 | 22.4 |
| InstructBLIP-7B | — | 49.2 | 34.5 | 60.5 | 50.1 | — | — | 36.0 | 23.7 | 53.4 | 60.9 | 26.2 |
| InstructBLIP-13B | — | 49.5 | 33.4 | 63.1 | 50.7 | 78.9 | 1212.8 | — | — | - | 58.2 | 25.6 |
| Shikra | 77.4* | — | — | — | — | — | — | 58.8 | — | - | — | — |
| IDEFICS-9B | 50.9 | 38.4 | 35.5 | — | 25.9 | — | — | 48.2 | 25.2 | - | — | — |
| IDEFICS-80B | 60.0 | 45.2 | 36.0 | — | 30.9 | — | — | 54.5 | 38.1 | - | — | — |
| Qwen-VL | 78.8* | 59.3* | 35.2 | 67.1 | 63.8 | — | — | 38.2 | 7.4 | 56.3 | — | — |
| Qwen-VL-Chat | 78.2* | 57.5* | 38.9 | 68.2 | 61.5 | — | 1487.5 | 60.6 | 56.7 | 58.2 | — | — |
| LLaVA-1.5-7B | 78.5* | 62.0* | 50.0 | 66.8 | 58.2 | 85.9 | 1510.7 | 64.3 | 58.3 | 58.6 | 63.4 | 30.5 |
| InternVL-Chat | 79.3* | **62.9*** | 52.5 | 66.2 | 57.0 | 86.4 | 1525.1 | 64.6 | 57.6 | 60.6 | 65.9 | 30.9 |
| VILA-7B | 79.9* | 62.3* | **57.8** | 68.2 | 64.4 | 85.5 | 1533.0 | 68.9 | 61.7 | 61.1 | 69.7 | 34.9 |
| LLaVA-NeXT-7B | **81.8*** | 64.2* | 57.6 | 70.1 | 64.9 | **86.5** | 1519.0 | 67.4 | — | — | **81.6** | **43.9** |
| Ours-7B | 81.2* | 61.7* | 57.0 | **91.8*** | 65.2 | 85.4 | **1602.3** | **76.5** | **75.4** | 65.6 | 72.1 | 41.3 |

## B.3 SFT EXPERIMENT

To further validate the effectiveness of our image-text interleaved pre-training, we followed the approach of LLaVA-1.5 (Liu et al., 2023d), MM1 (McKinzie et al., 2024), and InternVL-1.5 (Chen et al., 2024c) to collect approximately 3.3M SFT examples from a diverse set of datasets, as shown in Table 8. These datasets are formatted into the instruction-following format, the same as LLaVA-1.5. During SFT, we train the entire model, including the vision encoder, MLP projector, and LLM. We compare our model with state-of-the-art MLLMs, as presented in Table 7. The results demonstrate that our image-text interleaved pre-training significantly enhances the model's performance.

Table 8: **Summary of datasets used in the SFT experiment.** To further validate the effectiveness of our image-text interleaved pre-training, we followed the approach of LLaVA-1.5 (Liu et al., 2023d), MM1 (McKinzie et al., 2024), and InternVL-1.5 (Chen et al., 2024c) to collect approximately 3.3M SFT examples from a diverse set of datasets.

| Task | Dataset |
|---|---|
| Captioning | TextCaps (Sidorov et al., 2020), ShareGPT4V (Chen et al., 2023a) |
| General VQA | VQAv2 (Goyal et al., 2017), GQA (Hudson & Manning, 2019), OKVQA (Marino et al., 2019), VSR (Liu et al., 2023a), KVQA (Shah et al., 2019), A-OKVQA (Schwenk et al., 2022), ViQuAE (Lerner et al., 2022) |
| Science | AI2D (Kembhavi et al., 2016), ScienceQA (Lu et al., 2022a), TQA (Kembhavi et al., 2017) |
| Chart | ChartQA (Masry et al., 2022), MMC-Inst (Liu et al., 2023c), DVQA (Kafle et al., 2018), PlotQA (Methani et al., 2020), LRV-Instruction (Liu et al., 2023b) |
| Mathematics | GeoQA+ (Cao & Xiao, 2022), TabMWP (Lu et al., 2022b), MathQA (Yu et al., 2023a), CLEVR-Math/Super (Lindström & Abraham, 2022; Li et al., 2023e), Geometry3K (Lu et al., 2021) |
| OCR | OCRVQA (Mishra et al., 2019), InfoVQA (Mathew et al., 2022), TextVQA (Singh et al., 2019), ArT (Chng et al., 2019), COCO-Text (Veit et al., 2016), CTW (Yuan et al., 2019), LSVT (Sun et al., 2019), RCTW-17 (Shi et al., 2017), ReCTs (Zhang et al., 2019), SynthDoG (Kim et al., 2022), LLaVAR (Zhang et al., 2023), DocVQA (Clark & Gardner, 2018) |
| Grounding | RefCOCO/+/g (Yu et al., 2016; Mao et al., 2016), Visual Genome (Krishna et al., 2017) |
| Conversation | LLaVA-150K (Liu et al., 2023e), LVIS-Instruct4V (Wang et al., 2023a), ALLaVA (Chen et al., 2024a), Laion-GPT4V (LAION, 2023), TextOCR-GPT4V (Jimmycarter, 2023), SVIT (Zhao et al., 2023) |
| Text-only | OpenHermes2.5 (Teknium, 2023), Alpaca-GPT4 (Taori et al., 2023), ShareGPT (Zheng et al., 2024) |

## B.4 SUBSET CURATION

We further filter higher-quality documents from OmniCorpus-CC. We curate the subsets with the attributes in the meta-annotation introduced in Section 3.4, including: (1) The fluency, non-advertisement, pornography, politics, and toxicity probability of documents. (2) The aesthetic and punsafe probabilities, width and height, aspect ratio, file size, and repetition rates (Collision frequencies of phash and dhash across the entire corpus) of images. (3) The number of images and paragraphs in the document.

We adjust the thresholds to control quality and quantity, obtaining the six subsets of different scales. To compare the average qualities of the subsets, we sampled 1 million documents from each subset to train models. As is shown in Table 3, from "988M" to "200M", as the threshold becomes stricter, the model benefits from better data quality. However, when document number decreases further, the data diversity is compromised, leading to a decline in model performance.

## B.5 ANALYSIS ON POSITION STRATEGIES

We choose Open-Flamingo as the cross-attention baseline. The Flamingo designs a masking approach to limit the number of visual tokens that a certain text token sees, i.e., 'At a given text token, the model only cross-attends to the visual tokens corresponding to the last preceding image/video.' (Refer to Appendix A.1.3 and Figure 7 of the Flamingo paper.)

The retrieval-based method ensures maximum similarity between the image and its adjacent text paragraphs, which intuitively makes it more suitable for training cross-attention-based MLLMs using the masking approach. In the LLaVA-like methods, where all images are attended to equally, the retrieval-based method disrupts the original layout of the multimodal document, leading to misunderstandings and a decline in performance.

## C DETAILS OF THE DATA ENGINE

### C.1 ADVANTAGES OF OUR PIPELINE SEQUENCE

In this section, we aim to demonstrate that our pipeline sequence is the fastest. We assume we have 10,000 CPU resources, 3 Gbps bandwidth, and 1,000 GPU resources, and we observe that there are, on average, 2.97 images in a document. It is evident that we must perform main body extraction first and preliminary text filtering before detailed text filtering. So we define step ①: Preliminary Text Filtering, step ②: Document Deduplication with Text, step ③: Image Downloading & Filtering, step ④: Detailed Text Filtering. The detailed settings can be seen in Table 9. Since the main resource

cost in step ③ is bandwidth, it can be performed in parallel with other steps. Considering 1 billion documents, Table 10 shows the processing time for all scenarios, where the processes in parentheses indicate that they can be performed in parallel.

It can be observed from Table 10 that the order ①②④③ is the most efficient. Since we aim to preserve more diverse documents, we choose to perform ①②(③④), retaining all documents after ① and ② along with their filtering results ③ and ④.

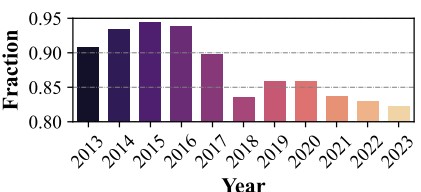

**Year**

Figure 7: **Trigger ratio of documents over years.** If a document is modified or filtered during our detailed text filtering, it will be included in the statistics.

Table 10: **Time to process 1B documents of different orders.** The processes in parentheses indicate that they can be performed in parallel. We find that ①②④③ is the optimal order, as changing any two steps would reduce the processing speed.

| Order | Time (hours) |
|---|---|
| ①②④③ | **2.31** |
| ①②(③④) | 5.95 |
| ①(③②)④ | 56.14 |
| (③①)②④ | 278.37 |
| ②①④③ | 2.65 |
| ②①(③④) | 6.30 |
| ②(③①)④ | 28.66 |
| (③②) ①④ | 278.26 |
| ①④②③ | 2.71 |
| ①④(③②) | 19.33 |
| ①(③④)② | 55.90 |
| (③①)④② | 279.59 |

Table 9: **Detailed settings of each step.** The processing speed and filtering ratio are calculated as averages in the real data pipeline.

| Step | #Doc/Second | Filtering ratio |
|---|---|---|
| ① | 1090k | 0.80 |
| ② | 388k | 0.90 |
| ③ | 3k | 0.40 |
| ④ | 100k | 0.67 |

## C.2 DETAILS OF THE HUMAN-FEEDBACK FILTERING

---

**Algorithm 1** Human Feedback Algorithm

---

**Require:** Documents $D^0 = \{d_1^0, d_2^0, ..., d_N^0\}$
**Ensure:** Filtering functions $F = \{f_1, f_2, ..., f_M\}$
1: $F \leftarrow \{\}$
2: **for** $i = 1$ to $step$ **do**
3:     Randomly sample $n$ documents $\hat{D}^{i-1} = \{d_1^{i-1}, d_2^{i-1}, ..., d_n^{i-1}\}$ from $D^{i-1}$
4:     Discovering $m$ problems by human feedback $P^i = \{p_1^i, p_2^i, ..., p_m^i\}$
5:     Generate $m$ filtering functions $F^i = \{f_1^i, f_2^i, ..., f_m^i\}$ according to $P^i$
6:     $F \leftarrow F + F^i$
7:     generate $D^i = \{d_1^i, d_2^i..., d_N^i\}$, where
8:     **for** each $d^i \in D^i$ **do**
9:         **for** each $f \in F^i$ **do**
10:             $d^i \leftarrow f(d^{i-1})$
11:         **end for**
12:     **end for**
13: **end for**

---

The overall algorithm for our human-feedback filtering is shown in Algorithm 1. We iteratively update the filtering function set several times based on human feedback to generate high-quality documents, such as those without unfinished paragraphs or social media information. The detailed functions and their corresponding false positive rates can be seen in Table 11. We sampled 1,000 documents to calculate the false positive rate. Many of these filtering functions have a false positive rate of zero, demonstrating the effectiveness of our designed filters. The trigger ratio of documents

for each year can be seen in Figure 7. We observe that our filtering functions work effectively across most documents, highlighting the necessity of our filters. Furthermore, we notice that the quality of documents in recent years is slightly better compared to older ones, resulting in a lower trigger ratio.

## C.3 DETAILS OF THE PRELIMINARY TEXT FILTERING

After extracting the original content from the website, we apply methods similar to C4 (Zhu et al., 2024) and Gopher (Rae et al., 2021) to eliminate extremely low-quality documents. Table 12 outlines the functions utilized during this stage.

Table 11: **Filtering rules.** The '-' indicates that the filtering function removed documents with hard indicators, rendering the false positive rate meaningless.

| Filter Function | False Positive Rate |
| --- | --- |
| *English Filtering Rules* | |
| Remove abnormal newlines in text | 0.0% |
| Split long underscores into paragraphs | 0.0% |
| Remove elements related to videos | 0.0% |
| Remove paragraphs with high number ratio | 0.0% |
| Remove keywords related to social media | 0.0% |
| Remove paragraphs with only one word | 0.0% |
| Remove very short paragraphs with keywords | 5.8% |
| Remove obviously aberrant list items | 0.0% |
| Remove citation and related content | 6.0% |
| Remove paragraphs ending with "readmore" | 0.0% |
| Remove incomplete sentences at ends | 16.7% |
| Remove video-related content | 0.0% |
| Remove URLs from text | 0.0% |
| Remove irrelevant image captions | 5.8% |
| Remove specific ads from domain | 0.0% |
| Mark articles with short paragraphs | 2.7% |
| Mark articles with lists and tables | 0.0% |
| Remove social media content | 2.1% |
| Remove overly short paragraphs | 8.3% |
| Remove paragraphs with many uppercase letters | 0.0% |
| Remove paragraphs with pornographic content | 0.0% |
| Remove footer content | - |
| Remove "like" and "follow" buttons | - |
| Remove short paragraphs | - |
| Remove paragraphs with word count issues | - |
| Remove documents with many non-letter words | - |
| Remove documents with few stop words | - |
| Remove documents with much pornographic content | - |
| Remove documents with bad paragraph length | - |
| *Chinese Filtering Rules* | |
| Remove duplicate lines and images | 4.0% |
| Remove source info like author, photographer | 10.0% |
| Remove sentences indicating newspaper flip | 0.0% |
| Remove lines matching keywords | 0.0% |
| Remove strange suffixes | 0.0% |
| Mark articles with empty images | 0.0% |
| Remove URLs from documents | 0.1% |
| Remove documents with low text-image ratio | 0.0% |
| Remove articles from cnnews-cepaper | 0.0% |
| Remove keywords related to videos | 0.0% |
| Fix empty titles from Baidu Baike | 0.0% |
| Fix list format errors from Baidu Baike | 0.0% |

| | |
|---|---|
| Remove recommendations and thanks to readers | 0.1% |
| Remove disclaimers and copyright statements | 0.1% |
| Remove content suspected of fraud | 0.1% |

Table 12: Filtering functions and thresholds for preliminary text filtering.

| Filter Function |
|---|
| *English Filtering Rules* |
| - Most characters of a document should be letters. |
| - The letters-to-numbers ratio should exceed 0.46. |
| - If the document contains more than 500 words, the most frequent word should not exceed 7.5%. |
| - If the document contains 500 words or fewer, the most frequent word should not exceed 30%. |
| - The document's word count should fall within the range [50, 100,000]. |
| - At least 80% of the words should contain at least one letter. |
| - The article should contain at least two stop words. |
| - The average word length should be between 3 and 10. |
| - Line number should be more than 3, and the third longest line should have at least 200 characters. |
| - The article should not contain "lorem ipsum". |
| - Find sentences that end with punctuation marks, extract the first and last lines, and remove the lines that are not within this range. |
| - Delete the lines containing phrases like "terms of use," "privacy policy,". |
| - Delete the lines with more than 1000 words. |
| *Chinese Filtering Rules* |
| - The document's word count should fall within the range [50, 100,000]. |
| - The character count must be greater than 150, with at least 30 Chinese characters, and the proportion of Chinese characters in the article should be no less than 60%. |
| - The article should contain at least two stop words. |
| - The article should have more than 3 lines, and the third longest line should have a length of at least 200 characters. |
| - Find sentences that end with punctuation marks, extract the first and last lines, and remove the lines that are not within this range. |
| - Delete the lines with more than 1000 words |

## C.4 DATA QUALITY ASSURANCE

A widely accepted view in the machine learning community is that data quality is more important than data quantity. As the OmniCorpus is the largest image-text interleaved dataset to date, we emphasize that the enormous data quantity primarily stems from the expansion of data sources and does not come with any compromise on data quality. The OmniCorpus-CC documents are processed from more available dumps in Common Crawl from 2013 to Nov./Dec. 2023 (while OBELICS (Laurençon et al., 2024a) process from Feb. 2020 to Jan./Feb. 2023). The OmniCorpus-CW is collected from major Chinese internet resources. The video sources of OmniCorpus-YT comprise YT-Temporal-1B (Zellers et al., 2022), HD-VILA-100M (Xue et al., 2022), HowTo100M (Miech et al., 2019), and InternVid (Wang et al., 2023c).

Besides, we strive to improve data quality by using a more strict filtering process than previous large-scale multimodal corpora. We emphasize that the human feed-back filtering is currently the most effective method for improving data quality significantly. The rules were iteratively refined to ensure that most unexpected content is filtered while the false positive rate are minimized. Hence, The data quality is ensured through substantial manual processing.

There are many decisions in the data engine that might affect the overall effectiveness of the dataset. Conducting exhaustive ablation studies on each threshold decision is highly resource-intensive, as adjusting the threshold for a single step requires re-running all subsequent steps and re-training the model. In this work, most of the thresholds were determined by manually reviewing documents within different value ranges. The thresholds were desired to remove most (>95%) low-quality documents while keeping the false positive rate low (<10%).

We have made substantial efforts to improve data quality while expanding the dataset. Validation metrics (see Section 4) and pre-training experiment results (see Section 5.3) also demonstrate the superior data quality. Due to its greater diversity, we encourage research on data curation that leverages our dataset as a resource to further improve quality.

# D SUPPLEMENTARY DATA ANALYSIS

## D.1 DEMONSTRATIVE EXAMPLES OF OMNICORPUS

We select two examples from OmniCorpus-CC as well as OmniCorpus-CW and one example from OmniCorpus-YT, as presented in Table 13, Table 14, and Table 15, respectively.

Table 13: Two demonstrative documents selected from OmniCorpus-CC.

---

*Example 1:*

Mother's Day is fast approaching. What better way to say 'i love you' to your Mum this year, by creating her this unique necklace, tailoring the fabrics, colours and beads all to your Mum's personal tastes.

Cut out your desired collar shape from a sturdy felt.

Choose a collection of clear acrylic stones in a selection of shapes. Cover them with a thin chiffon material, so you can still see the facets of the gems. Gather the fabric at the back of the gem and tack it together.

Sew the fabric covered stones onto your felt collar. Position them so that they sit slightly higher than the top edge of the collar to hide the felt.

Line up a string of multi-coloured beads made from precious stones along the bottom edge of the collar. Tack the string to the collar every 3 beads.

Fill in the gaps between the gems and beads with sew-on genuine crystal diamante stones in clasps.

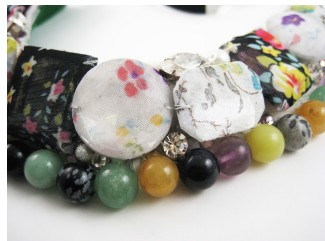

Measure a strip of black grosgrain ribbon to the length you wish your necklace to be. Cut it in half and stitch one end of each strip to the back of each tip to create the 'chain'.

Slot a ribbon end clasp onto the tip of each ribbon and close in place with a pair of jewellery pliers. Finish off with a screw clasp.

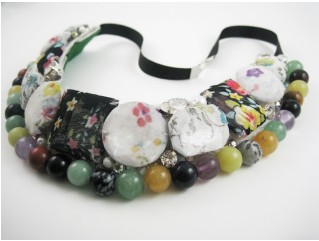

---

*Example 2:*

When my craft room came into being, at the end of February (actually it's still not missing the pink glass splashback..) I wanted the first thing I did to be something a bit special...

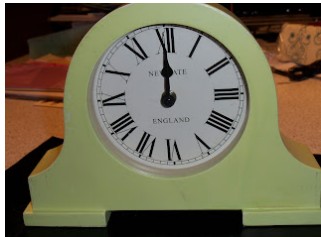

I found this clock on a clearance shelf, and whilst it was a bit in your face lime green, I liked the shape. I bought it, and put it to one side. Then I got inspiration...

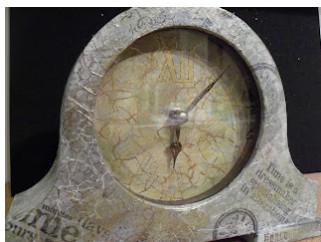

After a little bit of work, it now looks like this...

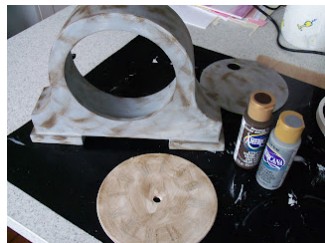

...and painted them up in decoart americana paint, roughly.

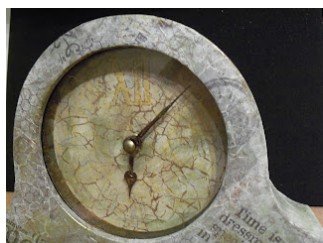

Putting it all together, the clock was sealed with claudine hellmuth multi medium, matte, which I also used as a 'glue' to cover the clock in the stamped tissue. I gave it another all over coat of the matte medium to seal it completely. There's also a smidge (or should I say smudges) of the grungold inka gold - it's so yummy! And now I have a really smart clock on my shelf!

Table 14: One demonstrative document was selected from OmniCorpus-YT.

***Example:***
Merry Christmas guys or happy Christmas. If you live in the UK, the marbles and I are going to show you what we got for Christmas.

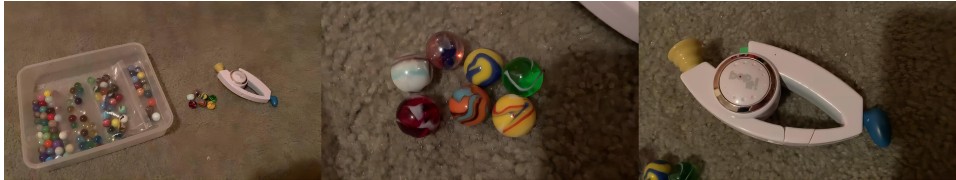

We have seven new rainbow marbles and the 2009 Bobbitt carabiner or carabiner. However it's pronounced yes this is new as you can see, and it was really cheap it was like twelve dollars yes. Anton told me on the note I wrote to him telling him what I want for Christmas and this works perfect.

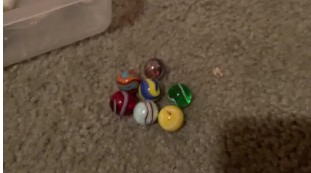

Anyways, moving on to the marbles. We have seven new rainbows.

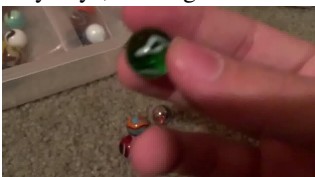

We have enchanted forest which is a clear green marble with white swirls.

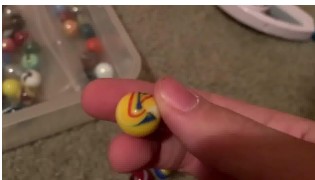

We have parrot which is a yellow marble with red blue and white swirls.

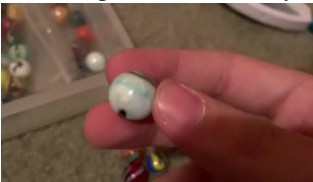

We have white tiger which is an IRA dies Dwight marble with blue and black swirls.

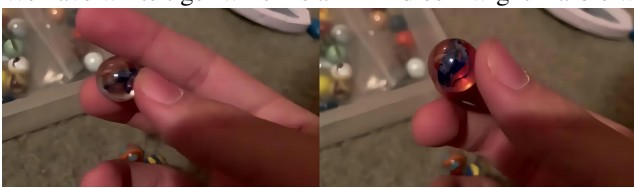

Next up, we have sunrise which is an IRA diced clear marble with a red and blue cat eye. It actually does look like a sunrise.

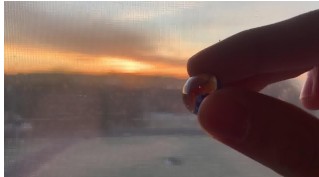

And if you guys look there is a beautiful sunrise outside of my house and here's sunrise right here does that not look like a sunrise. And yes, I still have my air conditioner ready anyways.

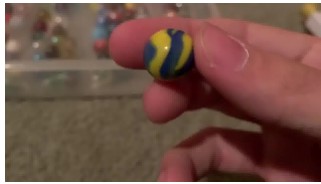

Next up, we have blue tang which is a yellow marble with blue swirls.

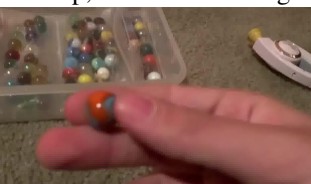

Next up, we have seahorse which is an orange marble with blue and black swirls.

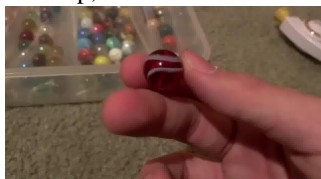

And last but not least, we have rooster which is a clear red marble with white swirls.

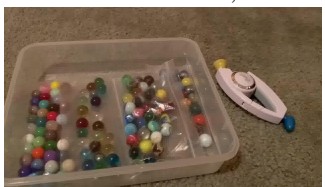

So that brings me to a total of 107 marbles soon to be 108 because apparently I got another marble but that's another house it's called Fiesta. I'll show you guys Fiesta once I'm in Florida alright guys.

I love you guys. Merry Christmas. And I'll see you guys on December 27 for the what we did in 2019 video. All right. I love you guys. Peace out.

Table 15: Two demonstrative documents selected from OmniCorpus-CW.

***Example 1:***
毫米波技术正广泛应用于无人驾驶

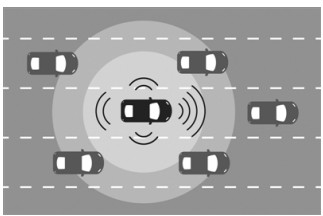

毫米波雷达指工作在毫米波波段的雷达，是测量被测物体相对距离、相对速度、方位的高精度传感器，早期被应用于军事领域，随着雷达技术的发展与进步，毫米波雷达传感器开始应用于汽车电子、无人机、智能交通等多个领域。

同超声波雷达相比，毫米波雷达具有体积小、质量轻和空间分辨率高的特点。与红外、激光、摄像头等光学传感器相比，毫米波雷达穿透雾、烟、灰尘的能力强，具有全天候全天时的特点。另外，毫米波雷达的抗干扰能力也优于其他车载传感器。由于毫米波在大气中衰减弱，所以可以实现更远距离的探测与感知，其中远距离雷达可以实现超过200米的感知与探测。

目前各个国家对车载毫米波雷达分配的频段各有不同，但主要集中在24GHz和77GHz。

频段在24GHz左右的毫米波雷达检测距离有限，因此常用于检测近处的障碍物，常被用来实现的功能有盲点检测、变道辅助等，主要为换道决策提供感知信息。

而性能良好的77GHz雷达的最大检测距离可以达到160米以上，因此常被安装在前保险杠上，正对汽车的行驶方向。长距离毫米波雷达能够用于实现紧急制动、高速公路跟车等功能；同时也能满足自动驾驶领域，对障碍物距离、速度和角度的测量需求。

---

***Example 2:***

三彩披鬃鞍马（唐）

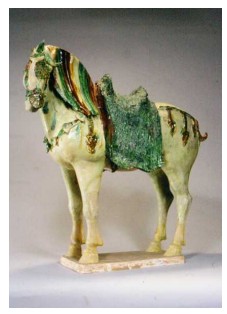

三彩披鬃鞍马，1990年陕西省西安市灞桥区半坡村出土，通高56.5cm，长58cm。

马首向左后方回望，两耳直竖，鬃毛左披，立于长方形踏板上。臀部发达，腿部强劲有力，特别是其眼睛、耳朵、筋骨、肌肉等局部雕琢精细，刀工娴熟，符合实体马的特点。马全身以白色为地釉，鬃毛为白、绿、褐三色相间；马鞍及垂于两侧腹下之毛织物为绿色釉；额前的当卢、耳鼻际的辔饰、胸前及尻上的革带及杏叶形垂饰均为黄、绿、褐三色釉；马尾为褐色。与一般唐三彩马相比，此马的釉色别具韵味，缺少大片鲜艳的红、黄、褐等色，而以素雅的白、绿色为主要色调，给人耳目一新的感觉。其造型俊洁、匀称，是唐三彩中罕见的精品。

唐代三彩马在造型上显示出宏大的气魄，体现着大唐王朝繁荣昌盛的景象，并从中可以看出唐人丰肥健壮的审美情趣。在形态上虽各有风采，但它们都有着共同的特征，即头小颈粗，臀圆背厚，四肢粗壮，而且骨肉匀停，线条流畅，内在的神韵在完美的造型中得到十足的体现，有力地烘托了盛世王朝的繁荣气象。

---

## D.2 STATISTICS

We follow Wanjuan-CC (Qiu et al., 2024) to compute several data quality metrics. As shown in Figure 8, a statistical analysis is conducted on various quantitative metrics of the documents, including document length, line count, token length, percentage of non-alphabetic characters, proportion of unique words, average word length, sentence count, stop-word ratio, and symbol-to-word ratio. The distributions enable the users to have a comprehensive understanding of the various characteristics of the data.

## D.3 TOPIC MODELING RESULTS

We follow previous works (Zhu et al., 2024; Laurençon et al., 2024a) to measure the diversity of the corpus with LDA (Blei et al., 2003), which presents the estimated proportions and related words for each topic. We train LDA with 20 topics on 100,000 documents randomly sampled from each partition of our dataset. The results on OmniCorpus-CC, OmniCorpus-YT, and OmniCorpus-CW are shown in Table 16, Table 17, and Table 18, respectively. Figure 5 shows T-SNE (Van der Maaten & Hinton, 2008) projection of LDA topic clusters. For each document, we generate a 20-dimensional vector and then reduce it to a 2-dimensional vector using T-SNE, allowing for visualization. From the Topic Modeling Result, we can find that the MMC4 and OmniCorpus-CC have several overlapping topics because they both originate from Common Crawl, and most topics are unique in all three sources, demonstrating the large diversity of the document content in OmniCorpus

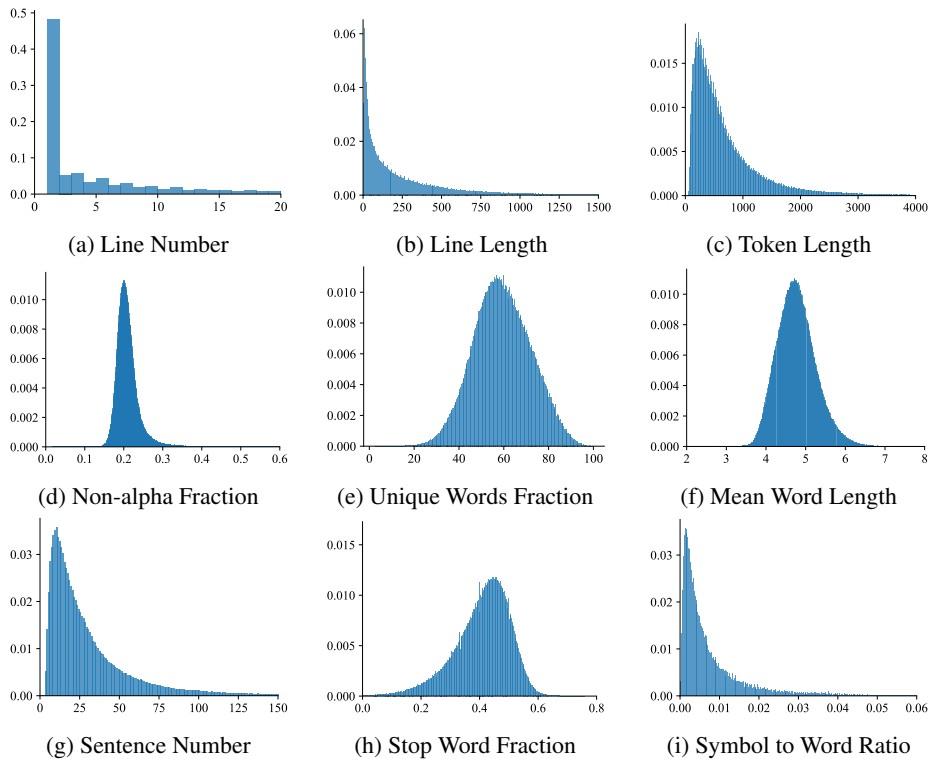

Figure 8: Percentage Statistics for Metrics on OmniCorpus-CC.

Table 16: The detailed topic modeling results of OmniCorpus-CC.

| Concept | Ratio | Related words |
|---|---|---|
| Legal System | 5.04% | court, men, march, prison, trial, media, department, national, wales, ceremony, issued, tower, medal, brexit, sheriff, service, gang, khan, charity, dead |
| Political Race | 3.21% | vote, ireland, mayor, taylor, elections, ford, usa, presidential, flag, seats, opposition, national, position, poll, circuit, lisa, dublin, colin, sox, eddie |
| Defense Tactics | 2.00% | security, photography, laura, miller, tennis, robin, douglas, norway, clarke, defence, rebels, anime, bryan, durham, grandpa, emperor, chad, manga, cia, rafael |
| Digital Analysis | 4.40% | trump, article, software, network, user, check, view, tool, method, device, mobile, biden, link, processing, output, resolution, document, audio, scale |
| Environmental Impact | 2.68% | paul, green, moon, waste, aircraft, toronto, wilson, bell, emissions, cemetery, tennessee, kentucky, flying, mars, crops, lawsuit, copper, belgium, idaho, violations |
| Educational Initiatives | 11.74% | program, service, national, department, schools, plan, quality, media, organization, world, teachers, campus, faculty, european, medicine, significant, innovation, clients, vision, journal |
| Creative Crafts | 4.25% | paper, card, items, cards, challenge, drawing, letters, settings, bags, craft, pen, custom, lesson, printed, knitting, tag, check, link, supplies, ribbon |
| Historical Europe | 6.55% | world, england, century, france, men, prayer, germany, queen, dead, stone, martin, die, italy, elizabeth, nations, view, castle, anna, powers, christians |

| Diverse Imagery | 4.09% | space, ice, wings, layout, stamp, angel, disabilities, ram, spain, mess, owl, satan, boot, header, flames, swap, unto, cracks, milwaukee, isaac |
|---|---|---|
| Familial Bonds | 6.35% | family, mother, young, christmas, boys, scotland, service, sam, mum, named, uncle, men, scottish, creep, picture, aunt, iowa, aged, rev, neighbors |
| Artistic Expression | 6.96% | music, world, films, audience, play, writers, actor, young, painting, visual, media, gallery, contemporary, steve, artistic, popular, reviews, entertainment, pop, poem |
| Cooking Essentials | 4.09% | cup, pan, milk, fresh, sauce, eggs, dish, chopped, meat, tsp, onion, salad, tbsp, tomatoes, raw, green, pasta, acid, fridge, corn |
| Weekend Fun | 9.33% | night, weekend, sunday, picture, saturday, cake, check, plan, hit, paint, cat, awesome, summer, bike, yeah, play, dead, view, wake, shoes |
| Cultural Variety | 2.14% | apple, foods, pakistan, davis, wire, pumpkin, mumbai, neil, hawaii, pearl, currency, louisiana, cluster, loaf, apples, moss, orleans, consultants, pudding, guild |
| Healthcare Solutions | 2.85% | skin, therapy, village, ukraine, drugs, ontario, immune, bath, tissue, lewis, ukrainian, twins, visa, infected, infections, substance, dose, respiratory, wifi, proteins |
| Seasonal Gardening | 4.11% | summer, garden, winter, spring, trees, soil, flowers, green, quilt, harvest, peppers, kenya, deer, fresh, ice, peas, planting, batch, microwave, abortion |
| Infrastructure Development | 5.48% | museum, construction, region, national, forest, airport, birds, parking, wood, vehicles, marine, village, service, view, lane, harris, trees, streets, electric, parts |
| Current Events | 2.28% | san, governor, com, ing, coronavirus, vaccine, bears, cable, luke, ter, rolls, bloom, cnn, moses, ash, lent, bees, chi, stitches, brunswick |
| Sports Thrills | 8.35% | game, play, players, club, football, sports, cup, winning, hit, saturday, score, scored, night, victory, competition, sunday, baseball, young, beat, ryan |
| Housing Market | 4.11% | price, housing, average, billion, cent, capital, period, smith, prices, estate, firm, numbers, hospitals, records, rose, commercial, significant, march, trend, latest |

Table 17: The detailed topic modeling results of OmniCorpus-YT.

| Concept | Ratio | Related words |
|---|---|---|
| Assembly Tools | 8.06% | side, ahead, bottom, half, tape, frame, size, slide, kit, add, tool, plug, wire, screw, holes, table, double, sides, screws, panel |
| Political Religion | 5.13% | president, government, jesus, war, state, lord, political, father, africa, international, christ, japan, donald, korea, may, minister, truth, foreign, pray, faith |
| Sports Competition | 6.10% | season, win, goal, league, teams, final, fans, half, round, score, competition, basketball, tonight, winner, sport, shot, plays, side, pitch, title |
| Family Routine | 8.40% | kids, morning, girls, parents, live, beautiful, birthday, hours, yesterday, saw, tonight, dance, bathroom, table, hear, tired, waiting, coffee, lunch, makes |
| Makeup Routine | 2.82% | skin, lip, apply, powder, blend, ahead, liquid, photo, brushes, lashes, mac, mascara, add, primer, coverage, routine, blending, vitamin, favorite, makes |

| | | |
|---|---|---|
| Printed Media | 2.58% | book, page, board, list, write, copy, printed, title, add, photo, printing, author, craft, mustang, compression, acrylic, washi, images, favorite, macros |
| Gender Education | 2.93% | women, class, students, training, learn, schools, golf, culture, teaching, campus, state, society, industry, events, arts, sexual, youth, local, gender, role |
| Vehicle Features | 3.63% | paint, rear, painting, side, window, fragrance, windows, hood, steering, roof, coat, storage, beautiful, transmission, horsepower, motorcycle, sport, trunk, honda, makes |
| Financial Investment | 5.50% | money, dollars, dollar, cost, worth, value, spend, cash, buying, tax, may, income, local, businesses, marketing, spending, investment, industry, live, interest |
| Learning Methods | 8.11% | may, question, learn, key, makes, negative, positive, ways, live, specific, computer, rather, value, results, add, function, search, creating, images, mobile |
| Medical Health | 2.82% | blood, cancer, may, medicine, healing, emily, pregnancy, symptoms, recovery, drugs, emergency, sheriff, tissue, oxygen, trial, healthy, bacteria, labor, southwest, appointment |
| Urban Affairs | 5.98% | city, police, morning, live, tonight, county, state, hours, bus, west, local, officer, california, valley, officers, parking, department, neighborhood, travel, clouds |
| Animal Care | 2.36% | animals, cat, cats, cage, madrid, deer, species, hunting, euros, soccer, rescue, pets, ski, pig, trap, lion, cow, zoo, mattress, aquarium |
| Physical Exercise | 4.02% | side, feet, leg, arm, lower, ground, shoulder, knee, flat, knees, roll, jump, core, valve, exhale, kick, swing, grip, twist, weight |
| Cooking Ingredients | 5.46% | add, cheese, half, sauce, coffee, egg, bowl, ingredients, ahead, tastes, pour, powder, potatoes, vegetables, wine, stir, beef, onion, bacon, teaspoon |
| Fashion Preferences | 5.76% | beautiful, favorite, size, pair, outfit, pants, pizza, comfortable, side, bottom, jeans, makes, leather, rose, saw, dollars, dollar, walmart, tag, halloween |
| Fitness Activities | 3.45% | weight, boat, workout, exercise, fishing, fat, training, protein, foods, calories, healthy, bait, fitness, half, morning, exercises, bass, squat, rope, ups |
| Music Performance | 9.46% | man, shot, hear, sound, saw, sounds, hell, record, live, shooting, yep, makes, kill, money, songs, guitar, nobody, album, laughter, hmm |
| Outdoor Gardening | 2.43% | garden, winter, trail, shoe, land, soil, ground, beautiful, yarn, feet, hike, mountains, double, seed, concrete, fence, stitches, bucket, half, seeds |
| Popular Entertainment | 4.99% | king, disney, john, favorite, scene, magic, shows, fans, films, batman, marvel, stars, deck, comic, artists, artist, role, war, ship, may |

Table 18: The detailed topic modeling results of OmniCorpus-CW. The original Chinese concepts and related words are translated into English.

| Concept | Ratio | Related words |
|---|---|---|
| Academic Preparation (学业准备) | 3.56% | divide, open, classmate, interview, apply, review, college entrance examination (GAOKAO), prize number, question, code, subject, vocabulary, retest, balcony, foundation, resume, memorize, internship, school, capability |

| | | |
|---|---|---|
| Medical Technology (医学科技) | 2.90% | medical treatment, surgery, illness, discovery, experiment, substance, laser, flight, measurement, include, capability, launch, grain, drone, tumor, rocket, cat, crowd, medication, appointment |
| Regional Affiliation (地域关联) | 4.92 % | Japan, Shanghai, Hong Kong, Hangzhou, Taiwan, Zhejiang, Nanjing, army, Guangdong, region, Jiangsu, Xinjiang, Chen (a surname), family, dynasty, mainland, Yunnan, Tibet, combat, Shanghai City |
| Global Initiatives (全球计划) | 4.33% | United States, epidemic, region, plan, India, including, president, Ukraine, billion US dollars, ten thousand US dollars, recovery, appointment, Nezha (a mythical figure in Chinese mythology), Biden, currency, Russia, blockchain, supplies, Bitcoin, Soviet Union |
| Grassroots Governance (基层治理) | 5.47% | community, deputy, party, rural, village, county, secretary, director, mu (a unit of area, equal to 1/15 hectare), the 20th National Congress, guarantee, implement, province, region, railway, previous year, within the territory, characteristic, comrade, activity |
| School Education (学校教育) | 5.31% | education, school, activity, college, teaching, teacher, parent, excellent, occupation, practice, primary school, capability, classmate, campus, sports, public welfare, association, classroom, student, characteristic |
| Emotional Expression (情感表达) | 15.40% | very, like, discover, feel, find, finish, the other party, spot, several, not want, slowly, fast, seems like, probably, buy, that kind, capability, habit, body, unable to |
| Smart Devices (智能设备) | 3.86% | mobile phone, baby, long, function, millimeter, charge, plant, wide, glass, machine, dual, screen, inch, Haier (a brand), material, select and purchase, indoor, Bluetooth, tool, leaf |
| Culinary Experience (美食体验) | 3.83% | decoration, tea, delicious, fruit, restaurant, taste, texture, put in, dish, store, catering, refrigerator, breakfast, flavor, white liquor, fragrant, ingredients, free, milk tea, guest room |
| Legal Violations (法律违规) | 4.91% | journalist, discover, apply, handle, spot, contract, store, illegal, clause, activity, supermarket, receive, express delivery, afternoon, crime, takeout, party involved, customer, death penalty, violate |
| Software Development (软件开发) | 5.52% | functionality, software, foundation, capability, advertisement, including, website, effect, model, very, version, stage, solve, search, free, define, memory, non, discover, targeted |
| Film & Sports (影视体育) | 4.06% | movie, director, leading actor, portray, team, champion, season, ball, match, point, athlete, league, player, participate, competition, direct, World Cup, win, act, Chinese language |
| Enterprise Management (企业管理) | 4.94% | enterprise, capability, employee, plan, including, register, supply chain, foundation, supervision, wisdom, solve, financing, activity, red ball, guarantee, intellectual property, high-quality, launch, Chloe (a name), stage |
| Creative Arts (创意艺术) | 4.17% | music, creation, author, novel, program, song, this book, poem, artist, melody, one piece, band, lyrics, piano, literature and art, a book, copywriting, provident fund, graphics card, hard drive |
| Body Care (身体护理) | 4.13% | skin, body, effect, human body, clean, absorb, traditional Chinese medicine, very, complexion, sleep, repair, synopsis, online novel, lose weight, weight, efficacy, massage, function, moisturize, serialized |
| Urban Attractions (城市景点) | 5.30% | decorate, appointment, located in, tourist, area, square, subway, discover, weather, activity, cultural relic, square meter, landscape, forest, public transportation, archaeology, ancient, underground, subway line, gate |

| Insurance Products (保险产品) | 2.25% | apple, insurance, Wuhan, guarantee, buy, beef, insurance company, fodder, lottery, milk powder, unearthed, critical illness insurance, premium, Zhao (a surname), fruit, claims settlement, sum insured, Qualcomm, Wuhan City, chattel |
| --- | --- | --- |
| Fashion Coordination (时尚搭配) | 9.24% | very, buy, like, match, color, good, feel, super, good-looking, match, clothes, effect, appearance, exquisite, high-end, color, hahaha, summer, clean, texture |
| Automotive Technology (汽车科技) | 2.46% | car model, driving, engine, fitness, Xi'an, new car, vehicle, charging, body, car manufacturer, rescue, recycling, store, version, driver, ideal, Toyota, mass production, torque, fire-fighting |
| Investment Analysis (投资分析) | 3.43% | fund, estimate, return, stock, billion US dollars, approximately, point, financing, valuation, interest rate, currency, market capitalization, analyst, same period, rebound, in, futures, buy, account, holding |

## D.4 TOP-LEVEL DOMAINS

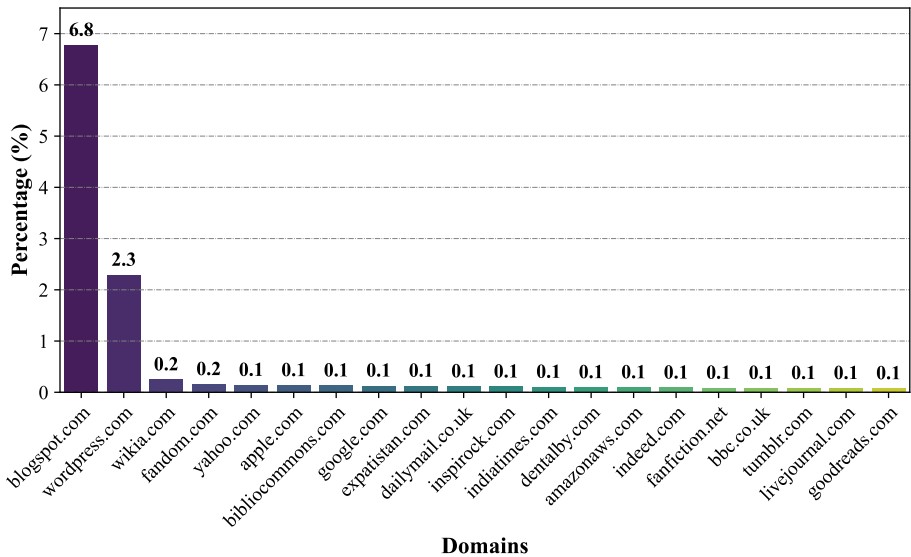

Figure 9: Top-20 Most Frequent Domains for Documents

We conduct an analysis of the top-level domains (TLDs) for the OmniCorpus-CCdataset. The documents are distributed across 16M domains. On average, each domain contains approximately 137 documents, with a median value of 4. As shown in Figure 9, the largest sources of documents are blogging platforms, accounting for nearly 9% of the total documents. Additionally, online encyclopedia platforms (e.g., Wikia), academic publication sites (e.g., BioRxiv), news media (e.g., Daily Mail and BBC), and e-commerce platforms (e.g., Amazon and Apple) are also prominent sources.

Images are distributed across 14 million domains, with each domain hosting an average of 615 images and a median of 6. Figure 10 shows that image sources are concentrated on a few major platforms, with Blogspot and WordPress accounting for over 10% of the total images. Cloud storage and content delivery networks (e.g., CloudFront and GoogleUserContent), shopping sites (e.g., Shopify and Amazon), and image hosting platforms (e.g., Flickr and Imgur) also hold significant shares. This high concentration indicates that users prefer using a few efficient platforms for uploading and sharing images, with cloud storage and content delivery networks playing a crucial role in image hosting.

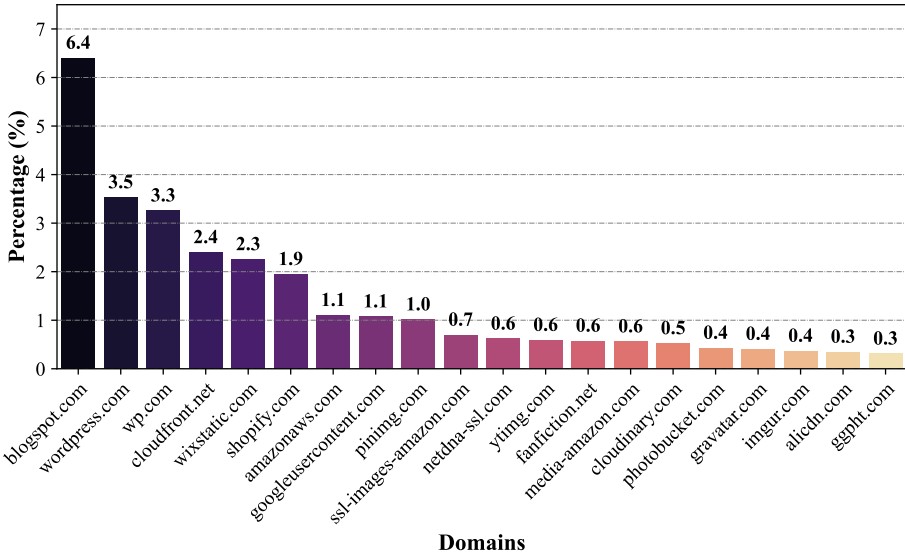

Figure 10: Top-20 Most Frequent Domains for Images

The OmniCorpus dataset shows that document sources are diverse, covering many fields and platforms, while image sources are concentrated, dominated by a few platforms. Blogging platforms are key for both, indicating their importance in user-generated content. The presence of online encyclopedias and academic sites underscores knowledge sharing, and the dominance of cloud storage highlights reliance on efficient services.

## E    AUTHOR STATEMENTS

The purpose of the statements is to clarify the responsibilities and liabilities associated with the project.

**Resource License:** The release and maintaining of the public resource are introduced in Section A.2. The OmniCorpus dataset is distributed under the CC BY 4.0 License[7], allowing users to copy, distribute, remix, adapt, and build upon the dataset. The open-source code is released under the Apache License 2.0[8], which allows users to use, modify, and distribute the software. This license also includes patent rights from contributors, protecting users from patent litigation.

**Legal Statement:** We have made extensive efforts to ensure the compliance and legality of our dataset collection process.

The data sources are carefully selected to minimize risks. OmniCorpus-CC is based on Common Crawl, a widely used resource in datasets like OBELICS (Laurençon et al., 2024a) and LAION (Schuhmann et al., 2022), which collects publicly accessible web content with inherently low risks of including sensitive personal information. OmniCorpus-CW is collected from the Chinese internet in full compliance with local laws and regulations. OmniCorpus-YT is derived from existing open video datasets, whose compliance with relevant legal standards has been verified in prior research. The dataset is outside the European Union, eliminating concerns related to General Data Protection Regulation (GDPR). Additionally, it is also fully complying with relevant laws of any country.

The authors assume full responsibility for the legal violations related to this project. We will continue to ensure that the dataset remains compliant with legal requirements and encourage users to adhere to relevant laws and ethical standards.

---

[7]https://creativecommons.org/licenses/by/4.0/
[8]https://www.apache.org/licenses/LICENSE-2.0

**Terms of Use:** To address compliance concerns, the Terms of Use (ToUs) have been developed based on widely accepted standards (such as those outlined for LAION (Schuhmann et al., 2022), MMC4 (Zhu et al., 2024), and OBELICS (Laurençon et al., 2024a)).

- All users, whether from academia or industry, must comply with the ToUs outlined in the CC BY 4.0 License.

- Any derived datasets or models must acknowledge the use of the OmniCorpus dataset to maintain transparency.

- The OmniCorpus must not be used in any project involving sensitive content or harmful outcomes, including but not limited to political manipulation, hate speech generation, misinformation propagation, or tasks that perpetuate harmful stereotypes or biases.

- The use of this dataset in any manner that violates rights, such as copyright infringement, privacy breaches, or misuse of sensitive information, is strictly prohibited.

- While we do not enforce jurisdiction-specific terms, we strongly recommend that users ensure compliance with applicable local laws and regulations.

- The use of specific subset must comply with the ToUs of the primary source. Specifically, the use of OmniCorpus-CC, OmniCorpus-CW, and OmniCorpus-YT must comply with the Common Crawl ToUs[9], the regulations on the security management of Internet data in China[10], and YouTube's ToUs[11], respectively.

- These ToUs do not supersede the ToUs of the original content sources. Users must ensure that any use of the dataset's content complies with the original ToUs and the rights of the data subjects.

By accessing or using this dataset, users acknowledge their responsibility to comply with all relevant legal, regulatory, and ethical standards. While the authors assume responsibility for the dataset's legal status, any legal consequences resulting from improper use of the dataset are the sole responsibility of the user.

Users who do not agree to the Terms of Use are not authorized to access or use the dataset. We will continue to make reasonable efforts to monitor and ensure the ethical and legal use of the dataset.

---

[9]https://commoncrawl.org/terms-of-use
[10]https://www.gov.cn/zhengce/content/202409/content_6977766.htm
[11]https://www.youtube.com/terms

