# OpenReview forum: "OmniCorpus: A Unified Multimodal Corpus of 10 Billion-Level Images Interleaved with Text"
_ICLR.cc/2025/Conference — ICLR 2025 Spotlight_

### Official Review · Reviewer_R6M4 · 2024-10-29

**Soundness:** 4
**Presentation:** 4
**Contribution:** 4
**Rating:** 8
**Confidence:** 5

**Summary:**

This manuscript introduces OmniCorpus, a massive multimodal dataset consisting of 10 billion-level images interleaved with text. This dataset is designed to support the development of multimodal large language models by providing a more diverse and larger scale of image-text data compared to existing datasets. The contributions of this manuscript include the introduction of the largest multimodal dataset, a set of tools and algorithms for data processing, and extensive experiments that validate the dataset's quality and effectiveness.  The authors conducted experiments to explore the effectiveness of image-text interleaved data for few-shot capabilities and language model maintenance. They also compared OmniCorpus with other datasets and found that their dataset outperforms others in terms of quality and diversity.

**Strengths:**

**Large Scale**: The dataset boasts an unprecedented scale of 8.6 billion images and 1.696 trillion text tokens, making it the largest multimodal dataset available.

**Diversity**: OmniCorpus includes data from a wide range of sources, including both English and non-English websites, as well as video-centric platforms, which enhances the diversity of the dataset.

**Usability**: The dataset has been validated through comprehensive analysis and experiments, demonstrating its quality, usability, and effectiveness.

**Writing**: This manuscript is well-written, with clear motivation and solid experimental discussions.

**Weaknesses:**

**Bias**: The paper acknowledges potential biases in the dataset but does not provide a detailed analysis of these biases.

**Filtering Mechanisms**: The current filtering process may not be sufficient to ensure high-quality data.

**Questions:**

Will all the data processing code and the data be open-sourced?

---

> ### Author Response · Authors · 2024-11-22
> **Feedback to Reviewer R6M4**
>
> **Q1: The paper acknowledges potential biases in the dataset but does not provide a detailed analysis of these biases.**
>
> **A1:** Collecting large-scale image-text data from the internet is widely regarded as a effective yet imperfect approach for scaling-up the large vision language model. While we have identified and filtered high-risk sources of bias, we acknowledge that potential biases still exist in the dataset. Similar to other web-crawled datasets (e.g., OBELICS[1], LAION[2], and MMC4[3]), the source data inherently contains systemic social biases (e.g., subpopulation depiction and racism), which are difficult to completely eliminate through automated processing.
>
> To analyze these potential biases, we computed the normalized Pointwise Mutual Information (nPMI) metric on 500k documents using the data-measurements-tool. The nPMI quantifies the association between terms, categories, or attributes, revealing potential biases by highlighting statistically significant relationships. For example, if certain word pairs have high nPMI scores, this suggests they frequently co-occur in the data, potentially reflecting stereotypes or biases.
>
> The results are uploaded to supplementary material (`bias_analysis` folder, we conduct calculation for twice, as in `npmi_1` folder and `npmi_2` folder). Given the extensive analysis results, we exhibit one biases analysis focused on gender-related content. We will make the detailed analysis results and analysising methodology available in our open-source repository, allowing users to review and consider them before use. Below, we focus on nPMI scores for terms associated with female (“she” and “woman”, in `combined-she-woman.md`) and male (“he” and “man”, in `combined-he-man.md`). The results indicate that male-associated terms are more frequently linked with topics like military, politics, and racing, while female-associated terms are often related to animals and domestic life.
>
> We encourage further research into bias analysis and mitigation strategies for internet-sourced image-text datasets. Additionally, we are committed to continuously maintaining the dataset and updating it with improved solutions as they become available.
>
> **Q2: Filtering Mechanisms: The current filtering process may not be sufficient to ensure high-quality data.**
>
> **A2:** We have spared no effort to ensure the data quality. As demonstrated by comparison results in Table 5, the quality of our documents are higher than the counterparts. As described in Appendix C.4 of the revised manuscript, we strive to improve data quality by using a more strict filtering process than previous large-scale multimodal corpora. The human feed-back filtering is currently the most effective method for improving data quality significantly. The rules were iteratively refined to ensure that most unexpected content is filtered while the false positive rate are minimized. Hence, The data quality is ensured through substantial manual processing. Additionally, it is flexible for the user to further enhance data quality by filtering based on metadata according to specific requirements.
>
> **Q3: Will all the data processing code and the data be open-sourced?**
>
> **A3:** Yes, as is described in '1 Introduction' section and 'A.2 Release And Maintaining' section. We follow common practices of dataset research, such as OBELICS[1], to release our work. We upload all the processed documents to public data hosting platforms. In additional to releasing data, we shall uphold the transparency in data collection and the reproducibility of model results. The developed human-feedback filtering functions and enhanced mainbody extraction tools will be available. The code for interleaved image-text pre-training with OmniCorpus, along with scripts for few-shot evaluation, will also be provided in the GitHub repository.
>
> **Reference:**
>
> [1] [Obelics: An open web-scale filtered dataset of interleaved image-text documents](https://proceedings.neurips.cc/paper_files/paper/2023/hash/e2cfb719f58585f779d0a4f9f07bd618-Abstract-Datasets_and_Benchmarks.html)
>
> [2] [LAION-5B: An open large-scale dataset for training next generation image-text models](https://proceedings.neurips.cc/paper_files/paper/2022/hash/a1859debfb3b59d094f3504d5ebb6c25-Abstract-Datasets_and_Benchmarks.html)
>
> [3] [Multimodal C4: An Open, Billion-scale Corpus of Images Interleaved with Text](https://proceedings.neurips.cc/paper_files/paper/2023/hash/1c6bed78d3813886d3d72595dbecb80b-Abstract-Datasets_and_Benchmarks.html)

---

### Official Review · Reviewer_n6DP · 2024-11-03

**Soundness:** 3
**Presentation:** 3
**Contribution:** 3
**Rating:** 8
**Confidence:** 3

**Summary:**

This paper introduces OmniCorpus, a 10 billion-level open-source image-text interleaved dataset. And it proposes an efficient data engine. It also conducts comprehensive analysis and experiments.

**Strengths:**

1. The largest open-source multimodal dataset to date. It pushes the boundaries of scale and diversity by encompassing 8.6 billion images interleaved with 1,696 text tokens.
2. A comprehensive set of tools and algorithms, including a streaming data format that unifies multimodal data from various sources, an efficient and scalable data engine capable of processing large-scale data, and human feedback filters to ensure high-quality data.
3. comprehensive analysis and experiments.

**Weaknesses:**

No obvious shortcomings observed.

**Questions:**

No obvious shortcomings observed.

---

> ### Author Response · Authors · 2024-11-22
> **Feedback to Reviewer n6DP**
>
> Thank you for your feedback. We appreciate your recognition of our work.

---

### Official Review · Reviewer_ff41 · 2024-11-04

**Soundness:** 3
**Presentation:** 3
**Contribution:** 3
**Rating:** 8
**Confidence:** 4

**Summary:**

The paper introduces a novel multi-modal and bilingual corpus at billions scale with image and text interleaved format. The dataset has three subsets (CC, CW, YT). The data processing and filtering steps for each subset is carefully designed and clearly explained. Extensive experiments are performed to access the both quality and the value of the dataset for multi-modal large language modeling tasks. Several  pre-training and fine-tuning experiments ablations demonstrate the value of the proposed dataset.

**Strengths:**

- Largest open source multi-modal dataset to date (8.6B images interleaved with text)
- Describes a detailed framework to collect and curate large multi-modal datasets at scale.
- Extensive ablations showing the value of the dataset with interleaved text format for few shot and other multi-modal understanding tasks.

**Weaknesses:**

- It is not 100% clear, if one can replicate the same data collection process and produce similar quality datasets. The models used for filtering content, the thresholds and potentially other important details seem to be missing.
- Table metrics and abbreviations need to be explained for clarity.

**Questions:**

A few quick notes for authors to improve the paper:
1) Table metrics and abbreviations should be clearly explained in the text where applicable.
2) minihash -> MinHash. We also need to understand what hash functions used to better understand how the dedup is performed for repeatability.
3) Table 5 - Shall the authors do a deep dive for the few shot evaluations with the proposed pre-training dataset (especially for COCO dataset)?  We need deeper understanding of why the few shot metric improvements are so high. Examples would also help especially for those the baseline models fail but the model pre-trained with OmniCorpus performs better.
4) Do the authors plan to release the code used for dataset generation?

**Details Of Ethics Concerns:**

There is no mention if the EU data is included or not (or going through a different treatment or not). If EU data is included, we need to make sure that the GDPR rules are followed. Not a legal expert here, but flagging this potential issue for visibility.

---

> ### Author Response · Authors · 2024-11-22
> **Feedback to Reviewer ff41 (1/2)**
>
> **Q1: Concern on missing details and thresholds of the data collection process.**
>
> **A1:** In the revised manuscript, we've included the missing details and thresholds:
>
> (1) We provide the source and thresholds of the models used for filtering in Section 3.1. We filtered out low-quality images using the LAION-Aesthetic Predictor[1] with a threshold of 3.7 and the LAION-NSFW Detector[1] with a threshold of 0.8. For detailed text filtering, we use several BERT[2] classifiers of WanJuan-CC[3] to score advertisement content, political content, toxic content, NSFW material, and document fluency.
>
> (2) We add a subsection named "preliminary text filtering" in Appendix C.3 of the revised manuscript, including the detailed description and thresholds of the filtering functions. For more details, please refer to the revised manuscript.
>
> We have added as many details as possible in the revised appendix. We summarize the descriptions here: Section 3 introduces the five key stages of the overall pipeline, the key improvements of our pipeline, the procedure of human-feedback filtering and the streaming data format. Appendix C.3 presents the descriptions and thresholds of the process in "preliminary text filtering" stage. Appendix C.2 present the description and false positive rate of the human-feedback filters in "detailed text filtering" stage. More details will also be availiable in our open-source repository.
>
> **Q2: Table metrics and abbreviations should be clearly explained in the text where applicable.**
>
> **A2:** Thanks for the suggestions. We have clarified the metrics and abbreviations in the revised manuscript.
>
> Specifically, in Table 1, "#" denotes "The number of". (modified in the caption of Table 1)
>
> In Table 2-4, "Avg. MLLM acc." means the mean value of the scores on OKVQA[4], TextVQA[5], COCO[6], and Flickr30k[7]. (added to the "Evaluation" paragraph of Section 5.2)
>
> In Table 2-5, "#Shot" means the number of in-context examples. (added to the "Evaluation" paragraph of Section 5.2)
>
> **Q3: minihash -> MinHash. We also need to understand what hash functions used to better understand how the dedup is performed for repeatability.**
>
> **A3:** Thanks, we'll fix this typo in the manuscript.
>
> We used minhash[8] to comparing the text content and remove the duplicate documents with a threshold of 0.8, which discarded approximately 90% of duplicates. We computed perceptual hash (phash) and difference hash (dhash) values of the images and remove the images that appeared more than 10 times across all the images.
>
> **Q4: A deep dive for the few shot evaluations.**
>
> **A4:** We upload comparison examples of our model and OpenFlamingo-9B[9] on 4-shot COCO[6] and OKVQA[4] dataset to supplementary material (few_shot_examples.pdf).
>
> These few-shot examples demonstrate that our model produces fewer hallucinations and more detailed descriptions in the image captioning task. Moreover, it provides more accurate and concise answers in the visual question-answering task. This validates that our dataset enhances the model’s contextual learning ability and text generation quality, which can be attributed to the higher quality and stronger contextual relevance of our documents.

---

> ### Author Response · Authors · 2024-11-22
> **Feedback to Reviewer ff41 (2/2)**
>
> **Q5: Do the authors plan to release the code used for dataset generation?**
>
> **A5:** Yes, as is described in '1 Introduction' section and 'A.2 Release And Maintaining'. In additional to releasing all the processed data, we shall release the code of processing English and Chinese documents, such as, the developed human-feedback filtering functions and enhanced mainbody extraction tools.
>
> **Q6: Concern on GDPR.**
>
> **A6:** We have made extensive efforts to ensure the compliance and legality of our dataset collection process.
>
> First, our data sources are carefully selected to minimize risks. OmniCorpus-CC is based on Common Crawl, a widely used resource in datasets like OBELICS[10] and LAION[1], which collects publicly accessible web content with inherently low risks of including sensitive personal information. OmniCorpus-CW comprises Chinese internet data sourced entirely outside the European Union, eliminating concerns related to GDPR and fully complying with relevant laws of any country and terms of use. OmniCorpus-YT is derived from existing open video datasets, whose compliance has been validated in prior works.
>
> Furthermore, we prioritize privacy by actively removing sensitive content, including personal identifiers, phone numbers, bank account details, email addresses, social media handles, and any content with opt-out signals. As is described in the datasheet in Appendix A.1.7, we shall delete specific samples that would be identified as sensitive.
>
> As a result, our dataset does not include EU data that raises GDPR concerns. Additionally, we will actively maintain the dataset to further enhance privacy and safety according to subsequent research and community feedback.
>
> **Reference:**
>
> [1] [LAION-5B: An open large-scale dataset for training next generation image-text models](https://proceedings.neurips.cc/paper_files/paper/2022/hash/a1859debfb3b59d094f3504d5ebb6c25-Abstract-Datasets_and_Benchmarks.html)
>
> [2] [Bert: Pre-training of deep bidirectional transformers for language understanding](https://eva.fing.edu.uy/pluginfile.php/524749/mod_folder/content/0/BERT%20Pre-training%20of%20Deep%20Bidirectional%20Transformers%20for%20Language%20Understanding.pdf)
>
> [3] [WanJuan-CC: A Safe and High-Quality Open-sourced English Webtext Dataset](https://arxiv.org/abs/2402.19282)
>
> [4] [OK-VQA: A Visual Question Answering Benchmark Requiring External Knowledge](https://openaccess.thecvf.com/content_CVPR_2019/html/Marino_OK-VQA_A_Visual_Question_Answering_Benchmark_Requiring_External_Knowledge_CVPR_2019_paper.html)
>
> [5] [Towards VQA Models That Can Read](https://openaccess.thecvf.com/content_CVPR_2019/html/Singh_Towards_VQA_Models_That_Can_Read_CVPR_2019_paper.html)
>
> [6] [Microsoft COCO Captions: Data Collection and Evaluation Server](https://arxiv.org/abs/1504.00325)
>
> [7] [Flickr30k Entities: Collecting Region-to-Phrase Correspondences for Richer Image-to-Sentence Models](https://openaccess.thecvf.com/content_iccv_2015/html/Plummer_Flickr30k_Entities_Collecting_ICCV_2015_paper.html)
>
> [8] [On the resemblance and containment of documents](https://ieeexplore.ieee.org/abstract/document/666900)
>
> [9] [OpenFlamingo: An Open-Source Framework for Training Large Autoregressive Vision-Language Models](https://arxiv.org/abs/2308.01390)
>
> [10] [Obelics: An open web-scale filtered dataset of interleaved image-text documents](https://proceedings.neurips.cc/paper_files/paper/2023/hash/e2cfb719f58585f779d0a4f9f07bd618-Abstract-Datasets_and_Benchmarks.html)

---

> ### Comment · Reviewer_ff41 · 2024-11-29
>
> The authors have addressed my questions with sufficient level of detail. Although I did not change the overall rating,  I increased the confidence level of the prior assessment.

---

### Official Review · Reviewer_M1Fb · 2024-11-06

**Soundness:** 2
**Presentation:** 3
**Contribution:** 3
**Rating:** 6
**Confidence:** 4

**Summary:**

The paper presents OmniCorpus, a large multimodal (text and vision) and multilingual (English and Chinese) dataset containing bilions of images interleaved with trilions of tokens. The paper explains how the data was obtained, filtered, and formated, and presents several experiments conducted on the dataset (i.e. training a VLM on the dataset from existing publicly-available vision encoders and text decoders).

**Strengths:**

- The paper presents OmniCorpus, a large multimodal (text and vision) and multilingual (English and Chinese) dataset containing hundreds of millions of documents (bilions of images, and trilions of tokens). This is by far the largest publicly available dataset that I know of, which can increase the amount of data available to conduct research on Vision-Language Models.
- The dataset has been carefully deduplicated and filtered to prevent NSFW content, personal information, offensive content, etc.

**Weaknesses:**

- As with many other datasets using crawled data from the Internet, it's not clear if 1) the authors of the paper themselves followed the terms of use of the sources of the data, and more importantly (from the user's perspective) if 2) the use of the downloaded data by the users (people training VLMs) may be subject to different terms of use / restrictions that are not directly stated anywhere, and may depend on different jurisdictions (e.g. can researchers legaly use this data to conduct research? both academic and industry researchers? can they release the models trained on this dataset?).
The authors acknowledge this in the "ethical discussion" in appendix A3 (and other parts of the appendix A): "it is impractical to obtain explicit consent from all content creators". I personally agree with this statement, but I think it should be mentioned in the main paper.
- I would appreciate a table similar to Table 5, but comparing the author's model train only on LAION (for instance) and OmniCorpus-CC, varying the number of the total number of tokens (e.g. text tokens + image tokens after encoding). This would be a proxy measuring the "quality" of both datasets, defined as "downstream accuracy that a token from the dataset provides". If the quality of the dataset proves to be relatively high, my soundness score would increase.
- It seems that the authors worked on improving the support of other languages beyond Chinese and English (e.g. line 237: "we [...] enhanced its capability to handle Chinese, Japanse and Arabaic documents"), however they decided to include only Chinese and English documents at the end. This is a lost opportunity (and amount of work) to have a truly multilingual (and not bilingual) dataset.
- As all the the other publicly available massive datasets, only the URL images are provided, which may difficult the reproducibility of the experiments conducted using it over time.

**Questions:**

- How was the set of "Chinese Websites" decided?
- I assume that the frequencies that appear in section 4 where obtained by manual inspection by the authors of the 200 randomly sampled documents, is that correct? or where they shipped to external evaluators?

---

> ### Author Response · Authors · 2024-11-22
> **Feedback to Reviewer M1Fb (1/2)**
>
> **Q1: Concern on Terms of Use.**
>
> **A1:** Thanks for your valuable comments regarding the compliance and ethical considerations of our dataset.
>
> In the revised manuscript, we have added clarifications regarding the terms of use to the 'Introduction' section.
>
> The collection of OmniCorpus was conducted with strict adherence to the terms of use (ToU) of the data sources. We followed established practices (e.g., OBELICS[1], LAION[2], and MMC4[3]) to exclude websites that prohibit data usage (e.g., employing the Spawning API to respect consent decisions). Furthermore, we applied additional manual filtering to exclude high-risk domains, especially for Chinese websites. For instance, although some platforms (e.g., online health question-answering sites) are publicly accessible, we excluded them due to potential ethical and privacy risks. Additionally, parts of the data source originate from existing datasets (e.g., we annotate text for videos from established datasets in OmniCorpus-YT) whose compliance has been rigorously validated in prior work.
>
> To address user concerns about compliance, we establish our ToU aligned with commonly accepted standards (such as ToU of OBELICS[1]). Users (whether from academia or industry) are expected to follow the [CC-BY ToU](https://creativecommons.org/terms/) and adhere to the ToU of the data sources (which are generally covered by the former). It is also required that any derived datasets or models should disclose the use of OmniCorpus for transparency. While we do not differentiate terms across jurisdictions, we encourage users to adapt the applicability of our dataset under their local legal frameworks.
>
> We appreciate your thoughtful suggestions, which have greatly contributed to improving the clarity and ethical transparency of our work. We will also update the manuscript and dataset repository to ensure that these guidelines are easily accessible to all users. Thank you again for helping us make OmniCorpus more robust and responsible.
>
> **Q2: Comparison experiment of LAION data and our data.**
>
> **A2:** Thank you for your suggestion. We conduct an experiment to compare LAION[2] and OmniCorpus-CC while aligning the total number of tokens.
>
> Specifically, we construct a LAION subset whose total valid tokens number aligns with that of the 1M OmniCorpus-CC subset. To be sound, only tokens within the maximum token length of the language model are considered valid. This process results in approximately 2.4M valid image-text pairs after filtering out images with invalid URLs or extremely small sizes (less than 10 pixels). We use the same LLaVA[4] architecture for pretraining on both subsets.
>
> The results are exhibited in the following table.
>
> |   #Shot    |    0     |    1     |    2     |    4     |
> | :--------: | :------: | :------: | :------: | :------: |
> | LAION only | **39.4** | **51.1** |   53.7   |   55.2   |
> | Ours only  |   28.4   |   48.3   | **54.4** | **58.7** |
>
> *(Each score represents the average of OKVQA[5]&TextVQA[6] accuracies and COCO[7]& Flickr30k[8] CIDEr[9] scores.)*
>
> The comparison reveals that each token from OmniCorpus-CC demonstrates superior accuracy in the 2&4-shot settings, while each token from the LAION subset exhibit better performance in the 0&1-shot settings.
>
> The superior performance of the LAION subset in the 0&1-shot settings can be attributed to its much higher image-to-text ratio. Since the LAION subset contains significantly more images compared to ours, it enhances performance in scenarios providing less context.
>
> In contrast, ours outperforms LAION subset in the 2&4-shot settings, highlighting the higher quality and in-context learning potential of our documents. The richer contextual associations in our native interleaved documents allow the model to better understand and utilize the provided examples, resulting in improved performance as the number of shots increases.
>
> We appreciate your suggestion, as it allowed us to demonstrate the strengths and trade-offs of both datasets more clearly. This comparison further underscores the value of OmniCorpus-CC for tasks that rely on in-context learning capabilities.

---

> > ### Comment · Reviewer_M1Fb · 2024-11-26
> >
> > **Q1**: Thank you very much for the clariftication. I think that the authors have done everything in their hands to provide the data under fair ToU.
> >
> > **Q2**: The experiment that you ran is not exactly what I asked for.
> >
> > You took a subset of LAION "whose total valid tokens number aligns with that of the 1M OmniCorpus-CC subset", trained a model in each subset, and then compared the quality of the two models under different few-shot scenarios (including zero-shot).
> >
> > However, these two datasets are way bigger than 1M. So how does the comparison look like if we use 10M-equivalent subsets? What about 100M or 1B? This sort of comparison is very interesting because it tells the potential user which dataset provides more bits/token under different training budgets. And most likely, the potential users are interested in the larger data regimes.
> >
> > I understand that this sort of experiment is quite compute-intensive, and might not be feasible to run it (I'll take that into account for my final decision), but it would highly influence the final score if one could show that OmniCorpus consistently outperforms LAION in terms of quality/token.

---

> ### Author Response · Authors · 2024-11-22
> **Feedback to Reviewer M1Fb (2/2)**
>
> **Q3: Lost opportunity for more language.**
>
> **A3:** In our project, human filtering is essential for quality assurance. Our data processing workers are currently only able to read and process Chinese and English, two of the most widely used languages. Hence, we process bilingual content for our dataset.
> The paragraph in question describes improvements to main body extraction, which indeed allows for more effective handling of multilingual documents. We hope to encourage researchers fluent in additional languages to further extend the filtering functions in future work. For this reason, we retained these extraction improvements to support further multilingual expansion.
>
> **Q4: About Image URL.**
>
> **A4:** Providing image files would significantly increase the cost of open-sourcing the dataset. Therefore, we follow common practices (e.g., OBELICS[1], LAION[2], and MMC4[3]) by representing images as URLs.
>
> **Q5: How was the set of "Chinese Websites" decided?**
>
> **A5:** We carefully selected Chinese websites from multiple legitimate and publicly accessible sources including platforms with clear Creative Commons agreements or similar open-content policies, public news media platforms, open Chinese article websites, and so on (e.g., Chinese Wikipedia, the Chinese Basic Corpus, and Chinese news platforms). We strictly avoided data with explicit restrictions on usage. This includes content containing personal information or privacy-related data (e.g., social media platforms like Weibo or online health question-answering sites) and content with strict copyright protections (e.g., CNKI or commercial databases). Our selection was guided by usage policies, ensuring all collected data will be legally and openly available for research purposes.
>
> **Q6: The qualitative assessment is obtained from whom?**
>
> **A6:** The qualitative assessment of the 200 samples was obtained by external evaluators.
>
> **Reference:**
>
> [1] [Obelics: An open web-scale filtered dataset of interleaved image-text documents](https://proceedings.neurips.cc/paper_files/paper/2023/hash/e2cfb719f58585f779d0a4f9f07bd618-Abstract-Datasets_and_Benchmarks.html)
>
> [2] [LAION-5B: An open large-scale dataset for training next generation image-text models](https://proceedings.neurips.cc/paper_files/paper/2022/hash/a1859debfb3b59d094f3504d5ebb6c25-Abstract-Datasets_and_Benchmarks.html)
>
> [3] [Multimodal C4: An Open, Billion-scale Corpus of Images Interleaved with Text](https://proceedings.neurips.cc/paper_files/paper/2023/hash/1c6bed78d3813886d3d72595dbecb80b-Abstract-Datasets_and_Benchmarks.html)
>
> [4] [Visual Instruction Tuning](https://proceedings.neurips.cc/paper_files/paper/2023/hash/6dcf277ea32ce3288914faf369fe6de0-Abstract-Conference.html)
>
> [5] [OK-VQA: A Visual Question Answering Benchmark Requiring External Knowledge](https://openaccess.thecvf.com/content_CVPR_2019/html/Marino_OK-VQA_A_Visual_Question_Answering_Benchmark_Requiring_External_Knowledge_CVPR_2019_paper.html)
>
> [6] [Towards VQA Models That Can Read](https://openaccess.thecvf.com/content_CVPR_2019/html/Singh_Towards_VQA_Models_That_Can_Read_CVPR_2019_paper.html)
>
> [7] [Microsoft COCO Captions: Data Collection and Evaluation Server](https://arxiv.org/abs/1504.00325)
>
> [8] [Flickr30k Entities: Collecting Region-to-Phrase Correspondences for Richer Image-to-Sentence Models](https://openaccess.thecvf.com/content_iccv_2015/html/Plummer_Flickr30k_Entities_Collecting_ICCV_2015_paper.html)
>
> [9] [CIDEr: Consensus-Based Image Description Evaluation](https://openaccess.thecvf.com/content_cvpr_2015/html/Vedantam_CIDEr_Consensus-Based_Image_2015_CVPR_paper.html)

---

> > ### Comment · Reviewer_M1Fb · 2024-11-26
> >
> > **Q3-Q6**: Thanks for your clarifications. They adequately addressed my concerns, so they are not a limiting factor for increasing my score.

---

> > > ### Author Response · Authors · 2024-12-01
> > > **Feedback to Reviewer M1Fb Round2 (1/3)**
> > >
> > > We sincerely appreciate your valuable and constructive reviews.
> > >
> > > In this response, we provide more implementation details and results of an additional experiment to strengthen the soundness.
> > >
> > > To make efficient use of the extended discussion period and potentially incorporate more data, we transitioned to a more computationally efficient model architecture, OpenFlamingo-3B[1].
> > >
> > > **Implement details:** We keep the average total number of tokens per batch to be consistent across datasets. Specifically, we first calculate the average effective total token count for the two datasets on OpenFlamingo-3B (using the function provided below). The inverse ratio of these values was used to determine the proportion of samples per batch (for rounding convenience in experiments, we allowed LAION-en-2B[2] to slightly exceed in token count). Finally, we adopt collect 2,048 OmniCorpus-CC documents or 22,016 LAION-en pairs for each step and train for 50k steps (equivalent to approximately 102M documents or 1.1B pairs), respectively. The model architecture utilizes CLIP ViT-L-14[3] and MPT-1B[4] as the backbone. The cross-attention interval is set to 1, and the learning rate is fixed at 1e-4[1]. Only the parameters of the cross-attention modules and the perceiver resampler are trainable, while all text embeddings (including the special tokens "<image>" and "<|endofchunk|>") are frozen. We set the warm-up step to 2,000. Furthermore, we leverage DeepSpeed ZeRO-1 and BF16 to accelerate training.
> > >
> > > ```python3
> > > def get_openflamingo_num_total_tokens(N_img: int, txt_list: list[str], max_tokens: int, tokenizer):
> > >     num_tokens_single_eoc = len(tokenizer.tokenize("<|endofchunk|>"))
> > >     num_tokens_single_image = len(tokenizer.tokenize("<image>"))
> > >     # each sentence is like "<bos>...<|endofchunk|><eos>"
> > >     num_bos_eos_eoc_tokens = 2 + num_tokens_single_eoc
> > >     num_total_image_related = (
> > >         # the first image only have one <image>
> > >         max(1, N_img) * num_tokens_single_image +
> > >         # the other images have <|endofchunk|><image>
> > >         max(0, N_img - 1) * (num_tokens_single_eoc + num_tokens_single_image)
> > >     )
> > >     num_total_text = sum([len(tokenizer.tokenize(t)) for t in txt_list])
> > >     return min(max_tokens, num_bos_eos_eoc_tokens + num_total_image_related + num_total_text)
> > > ```

---

> ### Author Response · Authors · 2024-12-01
> **Feedback to Reviewer M1Fb Round2 (2/3)**
>
> **Experimental results:** The table below summarizes the comparative results. In most cases, the model trained on OmniCorpus-CC outperforms the one trained on LAION-en-2B[2]. This performance gap is more pronounced compared to the 1M-scale experiment with LLaVA-1.5[5]. We attribute this to differences in model architecture and the foundational models' language capabilities. OpenFlamingo-3B[1], which employs MPT-1B[4], a relatively small-scale model, benefits significantly from the richer textual content in OmniCorpus-CC’s interleaved image-text data. In contrast, LLaVA-1.5-7B[5], built on Vicuna-1.5[6], already possesses strong contextual reasoning capabilities.  For LLaVA-1.5, effective alignment in the multimodal fine-tuning stage suffices to yield improvements, even with noisier datasets like LAION-en-2B.
>
> | Scale | data | f0 | f4 | f8 | c0 | c4 | c8 | o0 | o4 | o8 | t0 | t4 | t8 |
> | - | -: | -: | -: | -: | -: | -: | -: | -: | -: | -: | -: | -: | -: |
> | 500 steps / 250M tokens | LAION | 1.85 | 2.64 | 3.92 | 1.22 | 1.48 | 1.88 | 5.44 | 6.07 | 7.27 | 3.09 | 3.68 | 3.70 |
> | (1m docs / 11m pairs) | Ours | **2.66** | **3.69** | **4.62** | **2.53** | **4.84** | **5.65** | **12.40** | **14.01** | **14.68** | **6.60** | **6.93** | **7.27** |
> | 2500 steps / 1.25B tokens | LAION | 4.50 | 12.61 | **17.26** | 4.96 | 14.85 | 18.19 | 5.93 | 4.22 | 2.87 | 3.68 | 3.62 | 3.66 |
> | (5m docs / 55m pairs) | Ours | **9.01** | **13.24** | 13.06 | **14.47** | **29.79** | **28.67** | **15.91** | **16.63** | **17.15** | **6.60** | **6.90** | **7.05** |
> | 5k steps / 2.5B tokens | LAION | 3.92 | 15.58 | 20.07 | 5.18 | 20.06 | 26.46 | 5.57 | 5.21 | 4.20 | 4.91 | 4.39 | 4.88 |
> | (10m docs / 110m pairs) | Ours | **19.17** | **28.38** | **28.90** | **34.75** | **52.58** | **54.76** | **18.41** | **19.94** | **20.76** | **8.42** | **8.78** | **9.08** |
> | 10k steps / 5B tokens | LAION | 7.12 | 24.76 | 26.55 | 9.78 | 36.14 | 37.05 | 6.68 | 5.87 | 5.05 | 6.12 | 5.89 | 5.44 |
> | (20m docs / 220m pairs) | Ours | **26.50** | **37.68** | **37.35** | **39.25** | **63.39** | **67.84** | **21.73** | **24.41** | **24.46** | **12.01** | **12.63** | **12.43** |
> | 15k steps / 7.5B tokens | LAION | 5.83 | 21.97 | 24.63 | 10.05 | 26.51 | 24.23 | 6.57 | 4.63 | 2.98 | 7.31 | 4.83 | 4.60 |
> | (30m docs / 330m pairs) | Ours | **26.67** | **37.35** | **36.19** | **38.26** | **65.28** | **70.41** | **23.10** | **25.13** | **25.56** | **14.05** | **14.16** | **14.25** |
> | 20k steps / 10B tokens | LAION | 4.78 | 20.52 | 26.15 | 11.84 | 20.03 | 21.44 | 7.40 | 4.09 | 2.86 | 7.28 | 4.14 | 3.72 |
> | (40m docs / 440m pairs) | Ours | **28.68** | **35.57** | **40.12** | **44.02** | **58.02** | **67.83** | **21.02** | **23.60** | **25.56** | **14.35** | **15.51** | **15.70** |
> | 25k steps / 12.5B tokens | LAION | 9.31 | 26.93 | 27.16 | 16.25 | 28.62 | 24.94 | 7.93 | 4.91 | 4.60 | 8.91 | 4.62 | 4.14 |
> | (50m docs / 550m pairs) | Ours | **28.40** | **34.20** | **37.88** | **43.93** | **57.29** | **68.25** | **22.66** | **25.35** | **26.92** | **15.61** | **16.63** | **16.50** |
> | 30k steps / 15B tokens | LAION | 6.65 | 23.92 | 28.69 | 13.27 | 7.40 | 8.24 | 8.62 | 4.76 | 3.86 | 7.72 | 4.86 | 4.07 |
> | (60m docs / 660m pairs) | Ours | **26.83** | **37.76** | **42.04** | **45.05** | **64.63** | **72.08** | **24.18** | **26.20** | **27.59** | **16.24** | **17.02** | **17.09** |
> | 35k steps / 17.5B tokens | LAION | 5.85 | 24.71 | 27.08 | 11.49 | 16.88 | 4.75 | 8.17 | 5.62 | 5.72 | 9.08 | 4.22 | 5.12 |
> | (70m docs / 770m pairs) | Ours | **25.50** | **33.45** | **40.06** | **34.91** | **50.47** | **65.15** | **23.91** | **27.67** | **28.72** | **16.64** | **17.72** | **18.06** |
> | 40k steps / 20B tokens | LAION | 10.95 | 29.32 | 28.46 | 16.46 | 28.96 | 27.15 | 7.38 | 5.33 | 5.34 | 9.51 | 6.81 | 7.05 |
> | (80m docs / 880m pairs) | Ours | **24.00** | **36.51** | **43.27** | **38.33** | **58.67** | **72.24** | **24.64** | **26.97** | **28.75** | **16.44** | **17.80** | **17.10** |
> | 45ksteps / 22.5B tokens | LAION | 5.23 | 24.85 | 30.42 | 12.21 | 27.72 | 8.89 | 5.74 | 3.31 | 2.77 | 7.07 | 5.63 | 5.53 |
> | (90m docs / 0.99B pairs) | Ours | **33.24** | **35.42** | **44.07** | **49.75** | **47.99** | **55.10** | **25.21** | **27.33** | **28.97** | **15.95** | **18.08** | **18.41** |
> | 50k steps / 25B tokens | LAION | 9.82 | 29.33 | 28.84 | 7.59 | 13.42 | 3.09 | 4.98 | 4.74 | 4.87 | 8.06 | 5.11 | 5.50 |
> | (100m docs / 1,1B pairs) | Ours | **34.28** | **39.97** | **45.79** | **56.28** | **58.06** | **62.38** | **24.73** | **27.81** | **29.70** | **17.07** | **19.44** | **19.88** |
>
> ("f" indicates Flickr30k[7], "c" indicates COCO[8], "o" indicates OK-VQA[9], "t" indicates TextVQA[10].)

---

> ### Author Response · Authors · 2024-12-01
> **Feedback to Reviewer M1Fb Round2 (3/3)**
>
> Due to the significantly larger text token count of the nearly 1B-scale OmniCorpus-CC dataset compared to the entire LAION-2B, we were unable to complete the comparison at the 1B scale. For future experiments involving larger datasets and models, we kindly ask for the reviewers' understanding, as we have fully utilized the available resources during the extended discussion period. We have rigorously validated the data quality at multiple reasonable scales and believe that the conclusions drawn from these experiments are extensible to even larger datasets and models.
>
> **Reference:**
>
> [1] [OpenFlamingo: An Open-Source Framework for Training Large Autoregressive Vision-Language Models](https://arxiv.org/abs/2308.01390)
>
> [2] [LAION-5B: An open large-scale dataset for training next generation image-text models](https://proceedings.neurips.cc/paper_files/paper/2022/hash/a1859debfb3b59d094f3504d5ebb6c25-Abstract-Datasets_and_Benchmarks.html)
>
> [3] [Learning Transferable Visual Models From Natural Language Supervision](https://arxiv.org/abs/2103.00020)
>
> [4] [MPT-1B](https://huggingface.co/mosaicml/mpt-1b-redpajama-200b)
>
> [5] [Improved Baselines with Visual Instruction Tuning](https://arxiv.org/abs/2310.03744)
>
> [6] [Vicuna 1.5](https://huggingface.co/lmsys/vicuna-7b-v1.5)
>
> [7] [Flickr30k Entities: Collecting Region-to-Phrase Correspondences for Richer Image-to-Sentence Models](https://openaccess.thecvf.com/content_iccv_2015/html/Plummer_Flickr30k_Entities_Collecting_ICCV_2015_paper.html)
>
> [8] [Microsoft COCO Captions: Data Collection and Evaluation Server](https://arxiv.org/abs/1504.00325)
>
> [9] [OK-VQA: A Visual Question Answering Benchmark Requiring External Knowledge](https://openaccess.thecvf.com/content_CVPR_2019/html/Marino_OK-VQA_A_Visual_Question_Answering_Benchmark_Requiring_External_Knowledge_CVPR_2019_paper.html)
>
> [10] [Towards VQA Models That Can Read](https://openaccess.thecvf.com/content_CVPR_2019/html/Singh_Towards_VQA_Models_That_Can_Read_CVPR_2019_paper.html)

---

### Meta-Review · Area_Chair_Ldw7 · 2024-12-11

**Metareview:**

The paper proposes a very large dataset for MLLM research, including video. Its size, diversity and benchmarking form a significant contribution. Some concerns were raised about the ethics of the data collection and some aspects of the experiments, but seem well addressed. Three reviewers are strongly in favor of acceptance, and one weakly.

**Additional Comments On Reviewer Discussion:**

Reviewers engaged in discussion with the authors to a sufficient extent

---

### Decision · Program_Chairs · 2025-01-22

Accept (Spotlight)